# Effective Distributed Learning with Random Features: Improved Bounds and Algorithms

**Yong Liu**[1,2,*,†]**, Jiankun Liu**[3,*]**, Shuqiang Wang**[4]
[1]Gaoling School of Artificial Intelligence, Renmin University of China
[2]Beijing Key Laboratory of Big Data Management and Analysis Methods
[3]Institute of Information Engineering, Chinese Academy of Sciences
[4]Shenzhen Institutes of Advanced Technology, Chinese Academy of Sciences
 liuyonggsai@ruc.edu.cn, liujiankun@iie.ac.cn, sq.wang@siat.ac.cn

## Abstract

In this paper, we study the statistical properties of distributed kernel ridge regression together with random features (DKRR-RF), and obtain optimal generalization bounds under the basic setting, which can substantially relax the restriction on the number of local machines in the existing state-of-art bounds. Specifically, we first show that the simple combination of divide-and-conquer technique and random features can achieve the same statistical accuracy as the exact KRR in expectation requiring only $\mathcal{O}(|\mathcal{D}|)$ memory and $\mathcal{O}(|\mathcal{D}|^{1.5})$ time. Then, beyond the generalization bounds in expectation that demonstrate the average information for multiple trails, we derive generalization bounds in probability to capture the learning performance for a single trail. Finally, we propose an effective communication strategy to further improve the performance of DKRR-RF, and validate the theoretical bounds via numerical experiments.

## 1 Introduction

Kernel ridge regression (KRR) is one of the most popular nonparametric learning methods (Vapnik, 2000). Despite the excellent theoretical guarantees, KRR does not scale well in large scale settings because of high time and memory complexities (Liu et al., 2013; 2014; 2017; 2018; 2020b; Liu & Liao, 2015; Li et al., 2018; 2019c). Distributed learning (Zhang et al., 2013; Hsieh et al., 2014; Chang et al., 2017b; Li et al., 2019b; Lin et al., 2020), random features (Rahimi & Recht, 2007; Sutherland & Schneider, 2015; Rudi & Rosasco, 2017; Rudi et al., 2018; Liu et al., 2020a; Avron et al., 2017a; Yu et al., 2016; Jacot et al., 2020), and Nyström methods (Drineas & Mahoney, 2005; Ding & Liao, 2012; Yang et al., 2012; Camoriano et al., 2016; Si et al., 2016; Musco & Musco, 2017; Kriukova et al., 2017) are the most widely used large scale techniques to address the scalability issues. Recent statistical learning works on KRR together with large scale approaches demonstrate that these large scale approaches can not only obtain great computational gains but also can guarantee the optimal theoretical properties, such as KRR with divide-and-conquer (Zhang et al., 2013; 2015; Chang et al., 2017b;a; Guo et al., 2017; Lin et al., 2017; Li et al., 2019b;d; Lin et al., 2020), with random features (Rudi & Rosasco, 2017; Li et al., 2019e; Carratino et al., 2018; Yang et al., 2012), and with Nyström methods (Bach, 2013; Alaoui & Mahoney, 2015; Rudi et al., 2015; 2017; Ding et al., 2020).

The combinations of distributed learning and other large scale approaches are very intuitive but effective strategies to further improve the effectiveness, such as distributed learning with gradient descent algorithms (Lin & Zhou, 2018; Richards et al., 2020), with multi-pass SGD (Lin & Rosasco, 2017; Lin & Cevher, 2018; 2020), with random features (Li et al., 2019b), and with Nyström methods (Yin et al., 2020). The optimal generalization performance of these combining approaches has been studied, however, the main theoretical problem is that there is a strict restriction on the number of local machines. For sample, in (Lin & Zhou, 2018; Li et al., 2019b; Yin et al., 2020), to guarantee the optimal generalization performance in the basic setting, the upper bounds of the local machines are restricted to be a constant, which is difficult to be satisfied in real applications.

---

[*]Both authors contributed equally to this work
[†]Corresponding author

In this paper, we aim at enlarging the number of local machines by considering communications among different local machines. This paper makes the following three main contributions. Firstly, we improve the existing state-of-art results of the divide-and-conquer technique together with random features. We prove that the optimal generalization performance can be guaranteed even the partitions reach $\Omega(\sqrt{|\mathcal{D}|})$, which are limited to a constant $\Omega(1)$ for the existing bounds in the basic setting, $|\mathcal{D}|$ is the size of the data sets. Secondly, to essentially reflect the generalization performance, beyond the minimax optimal rates in expectation, we derive optimal learning rates in probability, which can capture the learning performance for a single trail. Finally, we develop a communication strategy to further improve the performance of our proposed method, and validate the effectiveness of the proposed communications via both theoretical assessments and numerical experiments.

**Related Work** The most related work includes the statistical analysis of distributed learning and random features.

**Distributed learning.** Optimal learning rates for divide-and-conquer KRR in expectation were established in the seminal work (Zhang et al., 2013; 2015). An improved bound was derived in (Lin et al., 2017) based on a novel tool of integral operator. Based on the proof techniques proposed in (Zhang et al., 2013; 2015; Lin et al., 2017), optimal learning rates were established for distributed spectral algorithms (Guo et al., 2017; Blanchard & Mücke, 2018; Lin & Cevher, 2020), distributed gradient descent algorithms (Lin & Zhou, 2018; Richards et al., 2020), distributed semi-supervised learning (Chang et al., 2017b), distributed local average regression (Chang et al., 2017a; Lin & Cevher, 2020), localized SVM (Meister & Steinwart, 2016), etc. Some other communication strategies for distributed learning have been provided, see e.g. (Fan et al., 2019; Li et al., 2019a; Lin & Cevher, 2020; Li et al., 2020), and references therein. The theoretical analysis mentioned above shows that the divide-and-conquer learning can achieve the same statistical accuracy as the exact KRR, however, there is a strict restriction on the number of local machines. The optimal learning rates with a less strict condition on the number of local machines for distributed stochastic gradient methods and spectral algorithms were established in (Lin & Cevher, 2020). In (Lin et al., 2020), they considered the communications among different local machines to enlarge the number of local machines. However, the communication strategy proposed in (Lin et al., 2020) based on an operator representation, which requires communicating the input data between each local machine. Thus, it is difficult to protect the data privacy of each local machine. Furthermore, for each iteration, the communication complexity of each local machine is $\mathcal{O}(|\mathcal{D}|d)$, where $d$ denotes the dimension, which is infeasible in practice for large scale data sets.

**Random Features.** The generalization bound of random features was first proposed in (Rahimi & Recht, 2008), which shows that $\mathcal{O}(|\mathcal{D}|)$ random features are needed for $\mathcal{O}(1/\sqrt{|\mathcal{D}|})$ learning rate. Some works further studied its theoretical performance (Cortes et al., 2010; Yang et al., 2012; Sutherland & Schneider, 2015; Sriperumbudur & Szabó, 2015; Bach, 2017; Avron et al., 2017b). By applying the standard integral operator framework (Smale & Zhou, 2007; Caponnetto & Vito, 2007), the optimal generalization bounds of KRR with random features were established in (Rudi & Rosasco, 2017), which requires only $\mathcal{O}(\sqrt{|\mathcal{D}|\log(|\mathcal{D}|)})$ random features. To decrease the size of random features, an improved approach was proposed based on a novel leverage score sampling strategy (Rudi et al., 2018). Sun et al. (2018) extended the result of (Rudi & Rosasco, 2017) to SVM. In (Li et al., 2019e), they further devised a simple framework for the unified analysis of random Fourier features, which can be applied to KRR, as well as SVM and logistic regression. To further improve the effectiveness, recently, Li et al. (2019b) considered the simple combination of divide-and-conquer and random features. However, to guarantee the optimal generalization performance, the number of local machines should be restricted to a constant, degenerating it into a single random features-based large scale KRR.

## 2  BACKGROUND

In a standard framework of supervised learning, there is a probability space $\mathcal{X} \times \mathcal{Y}$ with an unknown distribution $\rho$, where $\mathcal{X}$ is the input and $\mathcal{Y}$ is the output space. The sample set $\mathcal{D} = \{(\mathbf{x}_i, y_i)\}_{i=1}^n$ of size $n$ is drawn i.i.d from $\mathcal{X} \times \mathcal{Y}$ with respect to $\rho$. Let $K : \mathcal{X} \times \mathcal{X} \to \mathbb{R}$ be a Mercer kernel, and $\mathcal{H}_K$ be its reproducing kernel Hilbert space (RKHS) (Steinwart & Christmann, 2008; Vapnik, 2000), and assume that $K(\mathbf{x}, \mathbf{x}') \leq \kappa, \forall \mathbf{x}, \mathbf{x}' \in \mathcal{X}$. Throughout, we will denote the inner product in $\mathcal{H}_K$ by $\langle \cdot, \cdot \rangle_K$, and corresponding norm by $\| \cdot \|_K$.

**Kernel Ridge Regression (KRR)**

KRR is one of the most popular nonparametric learning methods (Shawe-Taylor & Cristianini, 2000; Vapnik, 2000), which can be stated as

$$f_{\mathcal{D},\lambda} = \underset{f \in \mathcal{H}_K}{\arg\min} \left\{ \frac{1}{|\mathcal{D}|} \sum_{i=1}^{|\mathcal{D}|} (f(\mathbf{x}_i) - y_i)^2 + \lambda \|f\|_K^2 \right\}, \tag{1}$$

where $\lambda > 0$ is the regularization parameter, $|\mathcal{D}|$ is the size of $\mathcal{D}$. Using the representation theorem (Shawe-Taylor & Cristianini, 2000; Vapnik, 2000), $f_{\mathcal{D},\lambda}$ can be written as $f_{\mathcal{D},\lambda}(\mathbf{x}) = \sum_{i=1}^{|\mathcal{D}|} \alpha_i K(\mathbf{x}_i, \mathbf{x})$ with $\boldsymbol{\alpha} = (\mathbf{K}_{\mathcal{D}} + \lambda \mathbf{I})^{-1} \mathbf{y}_{\mathcal{D}}$, where $\mathbf{K}_{\mathcal{D}} = \frac{1}{|\mathcal{D}|} [K_{\mathcal{D}}(\mathbf{x}_i, \mathbf{x}_j)]_{i,j=1}^{|\mathcal{D}|}$ is the $|\mathcal{D}| \times |\mathcal{D}|$ kernel matrix and $\mathbf{y}_{\mathcal{D}} = \frac{1}{|\mathcal{D}|}(y_1, \ldots, y_{|\mathcal{D}|})^{\mathrm{T}}$.

Despite the excellent theoretical guarantees (Blanchard & Krämer, 2010; Caponnetto & Vito, 2007), KRR requires $\mathcal{O}(|\mathcal{D}|^2)$ memory to store $\mathbf{K}_{\mathcal{D}}$, and $\mathcal{O}(|\mathcal{D}|^3)$ time to solve inverse of $\mathbf{K}_{\mathcal{D}} + \lambda \mathbf{I}$, which is infeasible for large scale settings.

**KRR with Random Features (KRR-RF)**

Assuming the spectral measure has a density function $\varrho(\cdot)$, the corresponding shift-invariant kernel can be written as $K(\mathbf{x}, \mathbf{x}') = \int_{\Omega} \psi(\mathbf{x}, \boldsymbol{\omega}) \psi(\mathbf{x}', \boldsymbol{\omega}) \varrho(\boldsymbol{\omega}) d\boldsymbol{\omega}$, where $\psi : \mathcal{X} \times \Omega \rightarrow \mathbb{R}$ is a continuous and bounded function with respect to $\boldsymbol{\omega}$ and $\mathbf{x}$. The main idea behind random Fourier features is to approximate the kernel function $K(\mathbf{x}, \mathbf{x}')$ by its Monte-Carlo estimation (Rahimi & Recht, 2007): $K_M(\mathbf{x}, \mathbf{x}') = \frac{1}{M} \sum_{i=1}^{M} \psi(\mathbf{x}, \boldsymbol{\omega}_i) \psi(\mathbf{x}', \boldsymbol{\omega}_i) = \langle \boldsymbol{\phi}_M(\mathbf{x}), \boldsymbol{\phi}_M(\mathbf{x}') \rangle$, where $\boldsymbol{\phi}_M(\mathbf{x}) = \frac{1}{\sqrt{M}} (\psi(\mathbf{x}, \boldsymbol{\omega}_1), \ldots, \psi(\mathbf{x}, \boldsymbol{\omega}_M))^{\mathrm{T}}$. The solution of KRR with random features can be written as

$$f_{M,\mathcal{D},\lambda}(\mathbf{x}) = \mathbf{w}_{M,\mathcal{D},\lambda}^{\mathrm{T}} \boldsymbol{\phi}_M(\mathbf{x}) \text{ with } \mathbf{w}_{M,\mathcal{D},\lambda} = (\boldsymbol{\Phi}_{M,\mathcal{D}} \boldsymbol{\Phi}_{M,\mathcal{D}}^{\mathrm{T}} + \lambda \mathbf{I})^{-1} \boldsymbol{\Phi}_{M,\mathcal{D}} \bar{\mathbf{y}}_{\mathcal{D}}, \tag{2}$$

where $\boldsymbol{\Phi}_{M,\mathcal{D}} = \frac{1}{\sqrt{|\mathcal{D}|}} (\boldsymbol{\phi}_M(\mathbf{x}_1), \ldots, \boldsymbol{\phi}_M(\mathbf{x}_{|\mathcal{D}|}))$ and $\bar{\mathbf{y}}_{\mathcal{D}} = \frac{1}{\sqrt{|\mathcal{D}|}}(y_1, \ldots, y_{|\mathcal{D}|})^{\mathrm{T}}$.

KRR-RF requires $\mathcal{O}(M|\mathcal{D}|)$ to store $\boldsymbol{\Phi}_{M,\mathcal{D}}$, $\mathcal{O}(M^3)$ and $\mathcal{O}(M^2|\mathcal{D}|)$ time to solve the inverse of $(\boldsymbol{\Phi}_{M,\mathcal{D}} \boldsymbol{\Phi}_{M,\mathcal{D}}^{\mathrm{T}} + \lambda \mathbf{I})$ and the matrix multiplication $\boldsymbol{\Phi}_{M,\mathcal{D}} \boldsymbol{\Phi}_{M,\mathcal{D}}$, respectively. Thus, the total space and time complexity of KRR-RF are $\mathcal{O}(M|\mathcal{D}|)$ and $\mathcal{O}(M^2|\mathcal{D}|)$, $M \ll |\mathcal{D}|$, respectively.

**Distributed KRR with Random Features (DKRR-RF)**

Let $\{\mathcal{D}_j\}_{j=1}^m$ be $m$ disjoint subsets with $\mathcal{D} = \cup_{j=1}^m \mathcal{D}_j$. The distributed KRR with random features (DKRR-RF) is defined as

$$\bar{f}_{M,\mathcal{D},\lambda}^0 = \sum_{j=1}^{m} \frac{|\mathcal{D}_j|}{|\mathcal{D}|} f_{M,\mathcal{D}_j,\lambda}, \tag{3}$$

where $f_{M,\mathcal{D}_j,\lambda}(\mathbf{x}) = \mathbf{w}_{M,\mathcal{D}_j,\lambda}^{\mathrm{T}} \boldsymbol{\phi}_M(\mathbf{x})$ with $\mathbf{w}_{M,\mathcal{D}_j,\lambda} = (\boldsymbol{\Phi}_{M,\mathcal{D}_j} \boldsymbol{\Phi}_{M,\mathcal{D}_j}^{\mathrm{T}} + \lambda \mathbf{I})^{-1} \boldsymbol{\Phi}_{M,\mathcal{D}_j} \bar{\mathbf{y}}_{\mathcal{D}_j}$. The space complexity, time complexity and communication complexity of DKRR-RF for each local machine are $\mathcal{O}(M|\mathcal{D}_j|)$, $\mathcal{O}(M^3 + M^2|\mathcal{D}_j|)$ and $\mathcal{O}(M)$, respectively.

## 3 DKRR-RF WITH COMMUNICATIONS (DRKK-RF-CM)

In this section, we will present an effective communication strategy to enlarge the number of local machines. We first give the motivation of our communication strategy, and then propose a communication-based method, called DRKK-RF-CM. The proposed communication strategy are adaptations from (Lin et al., 2020) to avoid communicating local data among partition nodes.

**Motivation.** Let $g_{M,\mathcal{D},\lambda} : \mathbb{R}^M \rightarrow \mathbb{R}^M$ be $g_{M,\mathcal{D},\lambda}(\mathbf{w}) := \left[ \boldsymbol{\Phi}_{M,\mathcal{D}} \boldsymbol{\Phi}_{M,\mathcal{D}}^{\mathrm{T}} + \lambda \mathbf{I} \right] \mathbf{w} - \boldsymbol{\Phi}_{M,\mathcal{D}} \bar{\mathbf{y}}_{\mathcal{D}}$. One can see that $2g_{M,\mathcal{D},\lambda}(\mathbf{w})$ is the gradient of the empirical risk of $\frac{1}{|\mathcal{D}|} \sum_{(\mathbf{x}_i,y_i) \in \mathcal{D}} \left( \mathbf{w}^{\mathrm{T}} \boldsymbol{\phi}_M(\mathbf{x}_i) - y_i \right)^2 + \lambda \|\mathbf{w}\|^2$ on $\mathbf{w}$. From Eq.2, we know that for any $\mathbf{w}$, the following equation holds:

$$\begin{aligned} \mathbf{w}_{M,\mathcal{D},\lambda} &= \mathbf{w} - \left[ \boldsymbol{\Phi}_{M,\mathcal{D}} \boldsymbol{\Phi}_{M,\mathcal{D}}^{\mathrm{T}} + \lambda \mathbf{I} \right]^{-1} \left[ \left[ \boldsymbol{\Phi}_{M,\mathcal{D}} \boldsymbol{\Phi}_{M,\mathcal{D}}^{\mathrm{T}} + \lambda \mathbf{I} \right] \mathbf{w} - \boldsymbol{\Phi}_{M,\mathcal{D}} \bar{\mathbf{y}}_{\mathcal{D}} \right] \\ &= \mathbf{w} - \left[ \boldsymbol{\Phi}_{M,\mathcal{D}} \boldsymbol{\Phi}_{M,\mathcal{D}}^{\mathrm{T}} + \lambda \mathbf{I} \right]^{-1} g_{M,\mathcal{D},\lambda}(\mathbf{w}). \end{aligned} \tag{4}$$

---

**Algorithm 1** Distributed KRR with Random Features and Communications (DKRR-RF-CM)

    **Initialize:** $\bar{\mathbf{w}}^0_{M,\mathcal{D},\lambda} = \mathbf{0}$

    **for** $t = 1$ **to** $p$ **do**

        **Local machine**: compute the local gradient $g_{M,\mathcal{D}_j,\lambda}(\bar{\mathbf{w}}^{t-1}_{M,\mathcal{D},\lambda})$, and communicate back to GM.

        **Global machine**: get the global gradient $g_{M,\mathcal{D},\lambda}(\bar{\mathbf{w}}^{t-1}_{M,\mathcal{D},\lambda}) = \sum_{j=1}^{m} \frac{|\mathcal{D}_j|}{|\mathcal{D}|} g_{M,\mathcal{D}_j,\lambda}(\bar{\mathbf{w}}^{t-1}_{M,\mathcal{D},\lambda})$ and communicate to each local machine.

        **Local machine**: compute $\boldsymbol{\beta}^{t-1}_j = \left[ \boldsymbol{\Phi}_{M,\mathcal{D}_j} \boldsymbol{\Phi}^{\mathrm{T}}_{M,\mathcal{D}_j} + \lambda \mathbf{I} \right]^{-1} g_{M,\mathcal{D},\lambda}(\bar{\mathbf{w}}^{t-1}_{M,\mathcal{D},\lambda})$ and communicate back to the global machine.

        **Global machine**: compute $\bar{\mathbf{w}}^t_{M,\mathcal{D},\lambda} = \bar{\mathbf{w}}^{t-1}_{M,\mathcal{D},\lambda} - \frac{|\mathcal{D}_j|}{|\mathcal{D}|} \sum_{j=1}^{m} \boldsymbol{\beta}^{t-1}_j$, and communicate to each local machine.

    **end for**

    **Output:** $\bar{\mathbf{w}}^p_{M,\mathcal{D},\lambda}$ **and** $\bar{f}^p_{M,\mathcal{D},\lambda} = \langle \bar{\mathbf{w}}^p_{M,\mathcal{D},\lambda}, \boldsymbol{\phi}_M(\cdot) \rangle$

---

Define $\bar{\mathbf{w}}^0_{M,\mathcal{D},\lambda} = \sum_{j=1}^{m} \frac{|\mathcal{D}_j|}{|\mathcal{D}|} \mathbf{w}_{M,\mathcal{D}_j,\lambda}$, it is easy to verify that

$$\bar{\mathbf{w}}^0_{M,\mathcal{D},\lambda} = \mathbf{w} - \sum_{j=1}^{m} \frac{|\mathcal{D}_j|}{|\mathcal{D}|} \left[ \boldsymbol{\Phi}_{M,\mathcal{D}_j} \boldsymbol{\Phi}^{\mathrm{T}}_{M,\mathcal{D}_j} + \lambda \mathbf{I} \right]^{-1} g_{M,\mathcal{D}_j,\lambda}(\mathbf{w}). \tag{5}$$

Comparing 4 and 5, and noting that the global gradient $g_{M,\mathcal{D},\lambda}(\mathbf{w})$ can be achieved via the communications of each local gradient $g_{M,\mathcal{D}_j,\lambda}(\mathbf{w})$, i.e., $g_{M,\mathcal{D},\lambda}(\mathbf{w}) = \sum_{j=1}^{m} \frac{|\mathcal{D}_j|}{|\mathcal{D}|} g_{M,\mathcal{D}_j,\lambda}(\mathbf{w})$, thus, we consider the following Newton Raphson iteration-based communication strategy:

$$\bar{\mathbf{w}}^t_{M,\mathcal{D},\lambda} = \bar{\mathbf{w}}^{t-1}_{M,\mathcal{D},\lambda} - \sum_{j=1}^{m} \frac{|\mathcal{D}_j|}{|\mathcal{D}|} \left[ \boldsymbol{\Phi}_{M,\mathcal{D}_j} \boldsymbol{\Phi}^{\mathrm{T}}_{M,\mathcal{D}_j} + \lambda \mathbf{I} \right]^{-1} g_{M,\mathcal{D},\lambda}(\bar{\mathbf{w}}^{t-1}_{M,\mathcal{D},\lambda}). \tag{6}$$

We propose an iterative procedure to implement the communication strategy Eq.6, which can be broken down into 4 steps. At first, each local machine computes the local gradient and communicates back to the global machine. Then the global machine computes the global gradient based on the local gradient, and communicates it to each local machine. In the third step, each local machine computes $\boldsymbol{\beta}^{t-1}_j$, and communicates back to the global machine. Finally, the global machine obtains the solution $\bar{\mathbf{w}}^t_{M,\mathcal{D},\lambda}$. More details can be seen in Algorithm 1.

**Complexity analysis.** Space complexity: each local machine only needs to store $\boldsymbol{\Phi}_{M,\mathcal{D}_j}$ and the local gradient $g_{M,\mathcal{D}_j,\lambda}$, thus the space complexity of each local machine is $\mathcal{O}(M|\mathcal{D}_j| + M) = \mathcal{O}(M|\mathcal{D}_j|)$; Time complexity: for each local machine, we only need to compute the matrix multiplication $\boldsymbol{\Phi}_{M,\mathcal{D}_j} \boldsymbol{\Phi}^{\mathrm{T}}_{M,\mathcal{D}_j}$ and the inverse of $\boldsymbol{\Phi}_{M,\mathcal{D}_j} \boldsymbol{\Phi}^{\mathrm{T}}_{M,\mathcal{D}_j} + \lambda \mathbf{I}$ once. For each iteration, we need to compute the local gradient $g_{M,\mathcal{D}_j,\lambda}$ and $\boldsymbol{\beta}_j$ for each local machine. Therefore, the total time complexity of each local machine is $\mathcal{O}(M^3 + M^2|\mathcal{D}_j| + pM|\mathcal{D}_j|)$, where $p$ is the number of communication; Communication complexity: for each iteration, we only communicate the local gradient $g_{M,\mathcal{D}_j,\lambda}$ and $\boldsymbol{\beta}_j$ to the global machine, and receive the gradient $g_{M,\mathcal{D},\lambda}$ and $\bar{\mathbf{w}}^{t-1}_{M,\mathcal{D},\lambda}$ from the global machine, so the total communication complexity is $\mathcal{O}(pM)$.

**Remark 1.** *From the complexity analysis above, we can see that if the number of the communication $p$ satisfying $p \leq M$ or $p \leq |\mathcal{D}_j|$, then the time and space complexity of DKRR-RF-CM are the same as DKRR-RF. Only the communication complexity is slightly increased from $\mathcal{O}(M)$ to $\mathcal{O}(pM)$.*

## 4 THEORETICAL ANALYSIS

In this section, we analyze the generalization performances of DKRR-RF and DKRR-RF-CM. The performance of the algorithm is usually measured by the expected risk $\mathcal{E}(f) = \int_{\mathcal{X} \times \mathcal{Y}} (f(\mathbf{x}) - y)^2 d\rho(\mathbf{x}, y)$. The optimal hypothesis $f_{\mathcal{H}_K}$ in $\mathcal{H}_K$ is denoted by $f_{\mathcal{H}_K} = \arg\min_{f \in \mathcal{H}_K} \mathcal{E}(f)$, and we assume $f_{\mathcal{H}_K}$ exists in the paper.

### 4.1 OPTIMAL LEARNING RATE FOR DKRR IN EXPECTATION

**Theorem 1.** *Suppose that $|\psi(\mathbf{x}, \boldsymbol{\omega})| \leq \tau$ almost surely, $\tau \in [1, \infty)$ and $|y| \leq \zeta$. If $\lambda = \Omega(|\mathcal{D}|^{-\frac{1}{2}})$, $|\mathcal{D}_1| = \ldots |\mathcal{D}_m|$, the number of partitions $m$ and the number of random features $M$ respectively correspond to $m \lesssim \sqrt{|\mathcal{D}|}$ and $M \gtrsim \sqrt{|\mathcal{D}|}$, then, for every $\delta \in (0, 1]$, with probability at least $1 - \delta$, we have $\mathbb{E}[\mathcal{E}(\bar{f}^0_{M,\mathcal{D},\lambda})] - \mathcal{E}(f_{\mathcal{H}_K}) = \mathcal{O}\left(|\mathcal{D}|^{-\frac{1}{2}} \log^2(1/\delta)\right).$*

From Theorem 1, one can see that if $m \lesssim \sqrt{|\mathcal{D}|}$ and $M \gtrsim \sqrt{|\mathcal{D}|}$, the learning rate of the generalization bound can reach $\mathcal{O}(1/\sqrt{|\mathcal{D}|})$, which is optimal in a minmax sense (Rudi & Rosasco, 2017; Caponnetto & Vito, 2007). It means that, in this basic setting, as long as the number of partitions and random features are in order $\Omega(\sqrt{|\mathcal{D}|})$, the corresponding ridge regression estimator has optimal generalization properties. The assumption of $|y| \leq \zeta$ can be related to the Bernstein condition or moment assumption (Blanchard & Krämer, 2016), but for simplicity, in this paper, we only consider that $\mathcal{Y}$ is bounded.

Optimal learning rates for divide-and-conquer KRR in expectation have been established in (Zhang et al., 2013; 2015; Lin et al., 2017), etc. However, there is a strict restriction on the number of local machines $m$. Specifically, in (Lin et al., 2017), to reach the optimal rate, $m$ should to restrict to a constant $m = \Omega(1)$. In (Li et al., 2019b), the authors have studied the generalization performance of the combination of divide-and-conquer technique and random features. Using the same setting as Theorem 1 (that is $r = 1/2$ and $\gamma = 1$ in Theorem 8 of Li et al. (2019b)), they prove that, if $M \gtrsim \sqrt{|\mathcal{D}|}$ and $m \lesssim \Omega(1)$, then $\mathbb{E}[\mathcal{E}(\bar{f}^0_{M,\mathcal{D},\lambda})] - \mathcal{E}(f_{\mathcal{H}_K}) = \mathcal{O}(|\mathcal{D}|^{-\frac{1}{2}} \log^2(1/\delta))$. It means that to guarantee the optimal generalization properties, the number of partitions should be restricted to a constant, but for our result is $\Omega(\sqrt{|\mathcal{D}|})$. In (Li et al., 2019b), they also considered using the unlabeled data to enlarge the number of partitions. They have proved that (see Corollary 12 of Li et al. (2019b) for detail), if $M \gtrsim \sqrt{|\mathcal{D}|}$ and $m \lesssim |\mathcal{D}^*|/|\mathcal{D}|$, then $\mathbb{E}[\mathcal{E}(\bar{f}^0_{M,\mathcal{D},\lambda})] - \mathcal{E}(f_{\mathcal{H}_K}) = \mathcal{O}(|\mathcal{D}|^{-\frac{1}{2}} \log^2(1/\delta))$, where $\mathcal{D}^*$ is the dataset includes both labeled and unlabeled data. Thus, if we want the number of partitions $m$ to reach $\Omega(\sqrt{|\mathcal{D}|})$ as the same as our Theorem 1, the size of $|\mathcal{D}^*|$ should be $\Omega(\sqrt{|\mathcal{D}|})$ times of $|\mathcal{D}|$. In this case, the data size of each local machine is $|\mathcal{D}| = |\mathcal{D}|^{3/2}/\sqrt{|\mathcal{D}|}$, so the time and space complexity are the same as KRR with a single random feature technique.

### 4.2 OPTIMAL LEARNING RATES FOR DKRR IN PROBABILITY

Note that $\mathbb{E}[\mathcal{E}(\bar{f}^0_{M,\mathcal{D},\lambda})] - \mathcal{E}(f_{\mathcal{H}_K}) = \mathbb{E}[\|\bar{f}^0_{M,\mathcal{D},\lambda} - f_{\mathcal{H}_K}\|^2_\rho]$ (Caponnetto & Vito, 2007), thus Theorem 1 proposes the optimal learning rate *in expectation*, which demonstrates the average information for multiple trails, but may fail to capture the learning performance for a single trail. To essentially reflect the generalization performance for a single trail, we derive the optimal learning rate *in probability*:

**Theorem 2.** *Under the same assumptions as Theorem 1. If $\lambda = \Omega(|\mathcal{D}|^{-\frac{1}{2}})$, $|\mathcal{D}_1| = \ldots |\mathcal{D}_m|$, $m \lesssim |\mathcal{D}|^{\frac{1}{4}}$ and $M \gtrsim |\mathcal{D}|^{\frac{1}{2}}$, then, for every $\delta \in (0, 1]$, with probability at least $1 - \delta$, we have $\left\|\bar{f}^0_{M,\mathcal{D},\lambda} - f_{\mathcal{H}_K}\right\|^2_\rho = \mathcal{O}\left(|\mathcal{D}|^{-\frac{1}{2}} \log^2(1/\delta)\right).$*

To guarantee the optimal generalization properties in probability, the number of partitions should be restricted to $\Omega(|\mathcal{D}|^{1/4})$, which is stricter than $\Omega(|\mathcal{D}|^{1/2})$ in Theorem 1. This is because the generalization error in expectation can be decomposed into approximation error, sample error and distributed error (more details can be seen in Proposition 1 in Appendix), but the error decomposition in probability is not easy to separate a distributed error in probability to control the number of local machines. The derive the optimal learning rate, we provide a novel decomposition, please see details in Proposition 9 in Appendix.

The following result demonstrates that the proposed communication strategy can enlarge the number of partitions in probability.

**Theorem 3.** *Under the same assumptions as Theorem 1, If $\lambda = \Omega(|\mathcal{D}|^{-\frac{1}{2}})$, $|\mathcal{D}_1| = \ldots |\mathcal{D}_m|$, $m \lesssim |\mathcal{D}|^{\frac{p+1}{2(p+2)}}$ and $M \gtrsim |\mathcal{D}|^{\frac{1}{2}}$, then, for every $\delta \in (0, 1]$, with probability at least $1 - \delta$, we have*

Table 1: Computational complexity required by different algorithms for the optimal learning rate $\mathcal{O}\big(1/\sqrt{|\mathcal{D}|}\big)$ in the basic setting. Logarithmic terms are not showed.

| Methods | Partitions $m$ | Random $M$ | Types | Space | Time | Communication |
|---|---|---|---|---|---|---|
| KRR (Caponnetto & Vito, 2007) | / | / | In probability | $|\mathcal{D}|^2$ | $|\mathcal{D}|^3$ | / |
| KRR-RF (Rudi & Rosasco, 2017) | / | $|\mathcal{D}|^{0.5}$ | In probability | $|\mathcal{D}|^{1.5}$ | $|\mathcal{D}|^2$ | / |
| KRR-Nyström Rudi et al. (2015) | / | $|\mathcal{D}|^{0.5}$ | In probability | $|\mathcal{D}|^{1.5}$ | $|\mathcal{D}|^2$ | / |
| DKRR (Zhang et al., 2015; Chang et al., 2017b) | $|\mathcal{D}|^{0.5}$ | / | In expectation | $|\mathcal{D}|$ | $|\mathcal{D}|^2$ | $|\mathcal{D}|^{0.5}$ |
| DKRR (Lin et al., 2020) | $|\mathcal{D}|^{0.25}$ | / | In probability | $|\mathcal{D}|^{1.5}$ | $|\mathcal{D}|^{2.25}$ | $|\mathcal{D}|^{0.75}$ |
| DKRR-CM (Lin et al., 2020) | $|\mathcal{D}|^{\frac{p+1}{2(p+2)}}$ | / | In probability | $|\mathcal{D}|^{\frac{p+3}{p+2}}$ | $|\mathcal{D}|^{\frac{3(p+3)}{2(p+2)}}$ | $pd|\mathcal{D}|$ |
| DKRR-RF (Li et al., 2019b) | $\Omega(1)$ | $|\mathcal{D}|^{0.5}$ | In expectation | $|\mathcal{D}|$ | $|\mathcal{D}|^2$ | $|\mathcal{D}|^{0.5}$ |
| **DKRR-RF (Theorem 1)** | $|\mathcal{D}|^{0.5}$ | $|\mathcal{D}|^{0.5}$ | In expectation | $|\mathcal{D}|$ | $|\mathcal{D}|^{1.5}$ | $|\mathcal{D}|^{0.5}$ |
| **DKRR-RF (Theorem 2)** | $|\mathcal{D}|^{0.25}$ | $|\mathcal{D}|^{0.5}$ | In probability | $|\mathcal{D}|^{1.25}$ | $|\mathcal{D}|^{1.75}$ | $|\mathcal{D}|^{0.5}$ |
| **DKRR-RF-CM (Theorem 3)** | $|\mathcal{D}|^{\frac{p+1}{2(p+2)}}$ | $|\mathcal{D}|^{0.5}$ | In probability | $|\mathcal{D}|^{\frac{2p+5}{2p+4}}$ | $|\mathcal{D}|^{\frac{3p+7}{2p+4}}$ | $p|\mathcal{D}|^{0.5}$ |

$$\left\| \bar{f}^{p}_{M,\mathcal{D},\lambda} - f_{\mathcal{H}_K} \right\|_{\rho}^2 = \mathcal{O}\left( |\mathcal{D}|^{-\frac{1}{2}} \log^{p+2}(1/\delta) \right), \text{ where } \bar{f}^{p}_{M,\mathcal{D},\lambda} \text{ is returned by Algorithm 1 under}$$

$p$-th iterations.

Compared Theorem 3 with Theorem 2, it is clear that the proposed communication strategy can relax the restriction on $m$ from $\Omega\left(|\mathcal{D}|^{1/4}\right)$ to $\Omega\left(|\mathcal{D}|^{(p+1)/(2(p+2))}\right)$. Note that $m$ is monotonically increasing with the number of communications $p$, which can demonstrate the power of the proposed communications. When $p \to \infty$, the partitions can reach $\Omega(\sqrt{|\mathcal{D}|})$, which is the same as the generalization bound in expectation.

**Remark 2.** *In the main text of this paper, we only give the optimal rates of DKRR-RF in the basic setting. The fast learning rates can be achieved under favorable conditions, see in Appendix.*

**Remark 3** (The Significance of Distributed Learning for RF). *At first glance, it seems that the bottleneck in learning with random Fourier features is not the size of the dataset but the number of features. However, from (Rudi & Rosasco, 2017), one can see that we only requiring $O(\sqrt{|\mathcal{D}|})$ random features to guarantee the optimal performance, so the total computational complexity is $\mathcal{O}(M^3 + M^2|\mathcal{D}|) = \mathcal{O}(|\mathcal{D}|^2)$. Thus, the computational bottleneck in learning is not only the size of random features, **but also the size of dataset**. If we don't consider reducing the size of $\mathcal{D}$, the computational complexity is $|\mathcal{D}|^2$ in the basic setting, which is not suitable for large scale problems. Distributed learning is one of the most popular methods to reduce the size of dataset. The distributed learning bring the distributed error, **but can decrease the variance** of the model Zhang et al. (2013; 2015). Thus, how to choose an appropriate number of partitions to trade off the distributed error and the variance to guarantee the optimal performance is a very interesting and significant direction.*

### 4.3 COMPARED WITH THE RELATED WORK

**Comparisons of the Time and Space Complexities**
Table 1 reports the statistical and computational properties of the related approaches and our theoretical findings under the basic setting. We see that our DKRR-RF can guarantee the optimal generalization performance in expectation only requiring $|\mathcal{D}|$ memory and $|\mathcal{D}|^{1.5}$ time, which is more effective than other methods. For DKRR-RF-CM, we can also see that it can guarantee optimal generalization performance in probability requiring less complexity than the communication-based method of DKRR-CM (Lin et al., 2020).

**Remark 4.** *In (Rudi et al., 2017), the authors considered combining the Nyström method and preconditioned conjugate gradient (PCG) (Cutajar et al., 2016) to scale up KRR. As far as we know, it is the only existing work that can guarantee optimal statistical accuracy, only requiring $|\mathcal{D}|$ memory and $|\mathcal{D}|^{1.5}$ time for KRR. In this paper, we consider combining distributed learning and random features, a completely different path from (Rudi et al., 2017). Note that in our proposed method, we need to compute the inverse of $\mathbf{\Phi}_{M,\mathcal{D}_j}\mathbf{\Phi}^{\mathrm{T}}_{M,\mathcal{D}_j} + \lambda\mathbf{I}$, which requires $|\mathcal{D}|^{1.5}$ time. Inspired by (Rudi et al., 2017), we can also adopt PCG to avoid the inverse calculation, which can further speed up our proposed DKRR-RF-CM. The combination of DKRR-RF-CM and PCG may open a path to reach the **linear time complexity** for optimal learning rate.*

**Novelty and Proof Techniques**

The most related works of our paper are (Li et al., 2019b), (Lin et al., 2020) and (Rudi & Rosasco, 2017). We discuss the novel techniques adopted to derive the improved results compared with them.

**Compared with (Li et al., 2019b).** (a) To derive the learning bounds, $\mathcal{Q}_{M,\mathcal{D}} := \|(C_M + \lambda I)^{-1/2}(C_M - C_{M,\mathcal{D}})(C_M + \lambda I)^{-1/2}\|$ is required to be estimated, where $C_M$ and $C_{M,\mathcal{D}}$ are self-adjoint and positive operators defined in Definition 1 (see in Appendix). In (Li et al., 2019b), they used a classical approach from (Chang et al., 2017b; Guo et al., 2017) to estimate $\mathcal{Q}_{M,\mathcal{D}}$ (see Lemmas 21 and 22 in (Li et al., 2019b)), and obtain that $\mathcal{Q}_{M,\mathcal{D}} \leq \frac{1}{\sqrt{\lambda}}\|(C_M - C_{M,\mathcal{D}})(C_M + \lambda I)^{-1/2}\| = \mathcal{O}(1/\lambda|\mathcal{D}| + \sqrt{\mathcal{N}(\lambda)/\lambda|\mathcal{D}|})$, where $\mathcal{N}(\lambda)$ is the effective dimension defined in Assumption 1 (see in Appendix). However, in our paper, we directly estimate $\mathcal{Q}_{M,\mathcal{D}}$ based on the concentration inequality for self-adjoint operators (Rudi & Rosasco, 2017; Lin & Cevher, 2020; 2018; Caponnetto & Yao, 2006), and prove that $\mathcal{Q}_{M,\mathcal{D}} = \mathcal{O}(1/(\lambda|\mathcal{D}|) + \sqrt{1/(\lambda|\mathcal{D}|)})$ (see in Proposition 6). Thus, our estimation of $\mathcal{Q}_{M,\mathcal{D}}$ is $\sqrt{\mathcal{N}(\lambda)}$ tighter than that in (Li et al., 2019b). This is the one of the key reasons why we can substantially relax the restriction on the number of local machines compared with (Li et al., 2019b); (b) We not only present the bounds in expectation, but also in probability. To derive the tight bound in probability, we provide new decompositions of $\|\bar{f}^0_{M,\mathcal{D},\lambda} - f_{M,\mathcal{D},\lambda}\|$ and $\|\bar{\mathbf{w}}^0_{M,\mathcal{D},\lambda} - \mathbf{w}_{M,\mathcal{D},\lambda}\|$, please see Proposition 9 in detail. As far as we know, these decompositions are novel; (c) We also consider the Newton Raphson iteration-based communication strategy. To derive the improved high-probability bounds with communication, we introduce a novel decomposition of $\|\bar{f}^t_{M,\mathcal{D},\lambda} - f_{M,\mathcal{D},\lambda}\|_\rho$ (see in Proposition 10).

**Compared with (Lin et al., 2020).** (a) In (Lin et al., 2020), they also considered a communication strategy to enlarge the number of local machines. At first it seems that it only need to communicate the gradient information, but it should be noted that the gradient information is based on an operator representation (see Eq.(7) in (Lin et al., 2020)), which is usually infeasible in practice. The authors present a realization for the proposed strategy by communicating the data among each local machine, see in Appendix B (page 34 in (Lin et al., 2020), step 1). Thus, the data privacy of each local machine is difficult to be protected. Furthermore, since it requires communicating data $\mathcal{D}_j$, $j = 1, \ldots, m$, among each local machine, for each iteration, the communication complexity of each local machine is $\mathcal{O}(|\mathcal{D}|d)$, which is too high for large scale data sets. However, the communication strategy proposed in this paper only requires communicating the gradient $g_{M,\mathcal{D}_j,\lambda}(\bar{\mathbf{w}}^{t-1}_{M,\mathcal{D},\lambda})$ and the model parameters $\beta^{t-1}_j$, rather than the data, therefore our proposed strategy do better on privacy protection. Moreover, the communication complexity is only $\mathcal{O}(M)$ for each local machine, $M \ll |\mathcal{D}|$, which is suitable for large scale data sets; (b) At first it seems that the proof techniques of (Lin et al., 2020) can be easily extended to our paper, but it is not true. If we use the same proof technical of (Lin et al., 2020), we can only obtain that $\|f^\diamond_{M,\mathcal{D},\lambda} - f_{M,\lambda}\|_\rho = \|(L_{M,\mathcal{D}} + \lambda I)^{-1}(L_{M,\mathcal{D}} - L_M)(f_\rho - f_{M,\lambda})\| = \mathcal{O}((1/(\sqrt{\lambda}|\mathcal{D}|) + \sqrt{\mathcal{N}(\lambda)/|\mathcal{D}|})\|f_\rho - f_{M,\lambda}\|_\rho/\sqrt{\lambda})$, where $f^\diamond_{M,\mathcal{D},\lambda}$, $f_{M,\lambda}$ and $f_\rho$ are defined in Definition 12. Combing with Proposition 2, one can only obtain that $\|f_{M,\mathcal{D},\lambda} - f_{M,\lambda}\|_\rho = \mathcal{O}((1/\lambda|\mathcal{D}| + \sqrt{\mathcal{N}(\lambda)/\lambda|\mathcal{D}|})\|f_\rho - f_{M,\lambda}\|_\rho)$. However, in our paper, we introduce new decompositions of $\|f^\diamond_{M,\mathcal{D},\lambda} - f_{M,\lambda}\|_\rho$ and $\|f_{M,\mathcal{D},\lambda} - f^\diamond_{M,\mathcal{D},\lambda}\|_\rho$ (see Propositions 2 and 3 in detail), and further obtain that $\|f_{M,\mathcal{D},\lambda} - f_{M,\lambda}\|_\rho = \mathcal{O}((1/(\sqrt{\lambda}|\mathcal{D}|) + \sqrt{\mathcal{N}(\lambda)/|\mathcal{D}|})\|f_\rho - f_{M,\lambda}\|_\rho)$ (see in Proposition 4), which is $\Omega(1/\sqrt{\lambda})$ tighter than the directly use of the techniques of (Lin et al., 2020). The novel decompositions $\|f^\diamond_{M,\mathcal{D},\lambda} - f_{M,\lambda}\|_\rho$ and $\|f_{M,\mathcal{D},\lambda} - f^\diamond_{M,\mathcal{D},\lambda}\|_\rho$ are the key reasons why we can guarantee the optimal performance even under $m = \Omega(\sqrt{|\mathcal{D}|})$. We only give an example here, but the novel decompositions have also been embedded in Propositions 5, 9, 10, etc.

**Compared with (Rudi & Rosasco, 2017).** (a) We study the statistical properties of the **combination** of distributed learning and random features, but in (Rudi & Rosasco, 2017), they only consider random features. As mentioned above, to estimate the tight bound of the distributed error, we introduce a novel decomposition of $\|\bar{f}^0_{M,\mathcal{D},\lambda} - f_{M,\mathcal{D},\lambda}\|$ to derive the tight bound in expectation, and novel decompositions of $\|\bar{f}^0_{M,\mathcal{D},\lambda} - f_{M,\mathcal{D},\lambda}\|$ and $\|\bar{f}^p_{M,\mathcal{D},\lambda} - f_{M,\mathcal{D},\lambda}\|$ to derive tight bounds in probability; (b) The combination not only brings distributed error, but also brings some other problems. If you compare the proofs of (Rudi & Rosasco, 2017) with these of our paper in detail, you can find that the decompositions of (Rudi & Rosasco, 2017) and ours are very different.

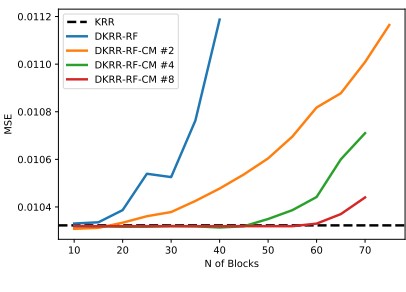 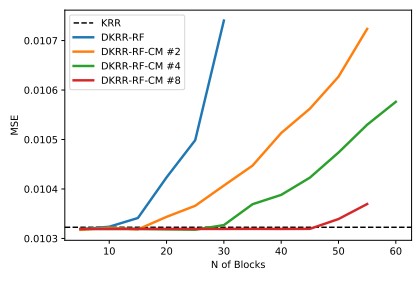

(a) simulated data, $q = 2$, $t = 6$        (b) simulated data, $q = 2$, $t = 5$

Figure 1: The mean square error or error rate on the test set with different partitions on KRR, DKRR-RF and our DKRR-RF-CM. # represents the number of communications.

Overall, we improve the existing state-of-art bounds in expectation, and provide novel communication-based distributed bounds with RF in probability. Moreover, we introduce some novel techniques and decompositions to substantially relax the restriction on the number of local machines, which are non-trivial extensions of (Li et al., 2019b; Lin et al., 2020; Rudi & Rosasco, 2017).

## 5 EXPERIMENTS

In this section, we validate our theoretical findings by performing experiments on both simulated and real datasets.

**Numerical Experiments.** Inspired by numerical experiments in (Rudi & Rosasco, 2017; Li et al., 2019e), we consider a spline kernel of order $q$: $K_{2q}(\mathbf{x}, \mathbf{x}') = 1 + \sum_{k=1}^{\infty} \cos(2\pi k(\mathbf{x} - \mathbf{x}'))/(k^{2q})$. If the marginal distribution of $\mathcal{X}$ is uniform on [0,1], then $K_{2q}(\mathbf{x}, \mathbf{x}') = \int_0^1 \psi(\mathbf{x}, \boldsymbol{\omega})\psi(\mathbf{x}, \boldsymbol{\omega})\varrho(\boldsymbol{\omega})d\boldsymbol{\omega}$, where $\psi(\mathbf{x}, \boldsymbol{\omega}) = K_q(\mathbf{x}, \boldsymbol{\omega})$ and $\varrho(\boldsymbol{\omega})$ is also uniform on [0,1]. The random features of the spline kernel are $\boldsymbol{\phi}_M(\mathbf{x}) = (\psi(\mathbf{x}, \boldsymbol{\omega}_1), \ldots, \psi(\mathbf{x}, \boldsymbol{\omega}_M))^{\mathrm{T}}/\sqrt{M}$. According to Theorem 1, 2 and 3, we set the size of the random features to be $M = \sqrt{|\mathcal{D}|}$, and fine tune $\lambda$ around $|\mathcal{D}|^{-1/2}$ using 5-fold cross validation[1], the tuned set is $\{2^{-5}, 2^{-3}, \ldots 2^5\}|\mathcal{D}|^{-1/2}$. We let the target function $f_*$ be a Gaussian random variable with mean $\mu = K_t(x, 0)$ and variance $\sigma^2 = 0.01$.

We generate 10000 samples for training and 10000 samples for testing. We use the exact KRR as a baseline, which trains all samples in a batch. We compare our proposed DKRR-RF-CM ($p = 2, 4, 8$) with KRR and DKRR-RF. We repeat the training 5 times and estimate the averaged error on testing data. The mean square error on the test set with different partitions is given in Figure 1(a, b), which can be summarized as follows: 1) When $m$ is not too large, the distributed methods (DRKK-RF and DRKK-RF-CM) are always comparable to original KRR. There exists an upper bound of $m$, when larger than it, the error increases dramatically and is far from the original KRR. This verifies the theoretical statement in Theorem 1, 2 and 3; 2) The upper bound $m$ of DKRR-RF-CM is much larger than DKRR-RF. This result is aligned with Theorem 3, which demonstrates that the proposed communication strategy can enlarge the upper bound $m$; 3) The upper bound $m$ of DKRR-RF-CM monotonically increases with the number of communications, which verifies Theorem 3.

**Real Data.** In this experiment, we consider the performance on real data. We use 6 publicly available datasets from LIBSVM Data[2]. The empirical evaluations with Gaussian kernel, $\exp(-\|\mathbf{x} - \mathbf{x}'\|^2/\sigma)$, are given in Figure 2, where the optimal $\sigma$ and $\lambda$ are selected by 5-fold cross-validation, $\sigma \in \{2^i, i = -10, -8, \ldots, 10\}$, $\{2^{-5}, 2^{-3}, \ldots 2^5\}|\mathcal{D}|^{-1/2}$, and the number of random features is $2\sqrt{|\mathcal{D}|}$.

---

[1]Because of the selection of the optimal $\lambda$ using the 5-fold cross validation, the computational complexity should be enlarged. However, it should be noted that even for the plain methods, tuning the optimal $\lambda$ is also required, which will enlarge the computational complexity as well.

[2]http://www.csie.ntu.edu.tw/~cjlin/libsvm.

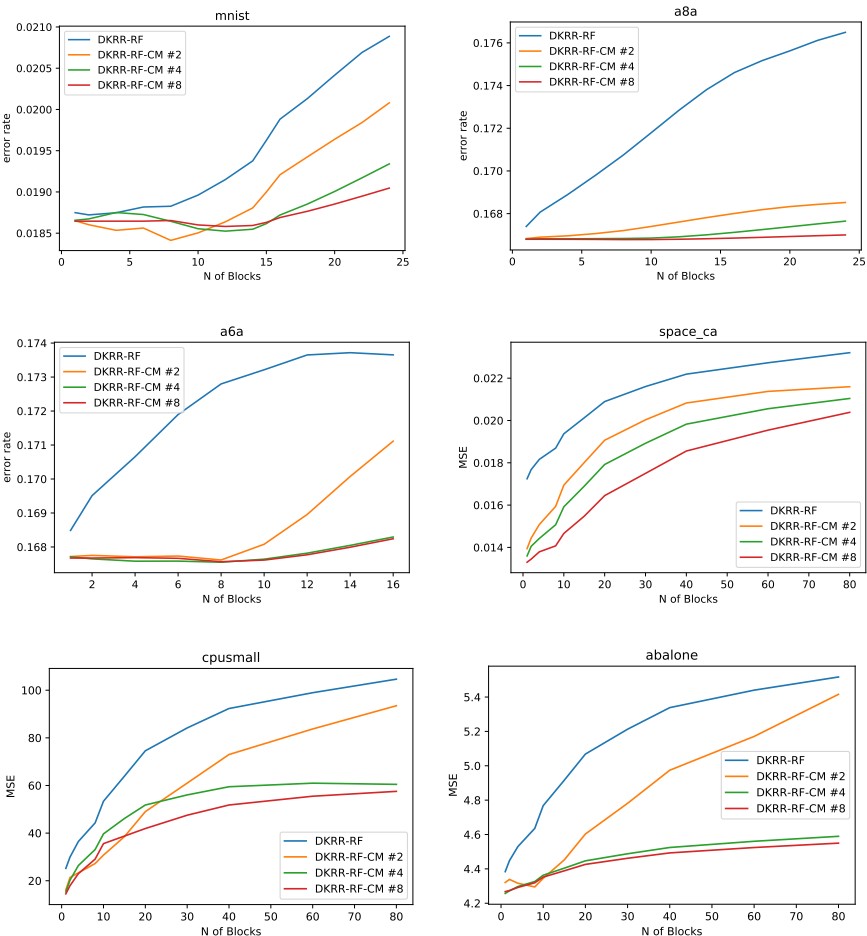

Figure 2: The mean square error or error rate on the test set with different partitions DKRR-RF and our DKRR-RF-CM on minist, a8a, a6a, space-ca, cpusmall and abalone. # represents the number of communications.

From Figure 2, one can find that: 1) our DKRR-RF-CM are better than the original DKRR-RF on all data-sets; 2) the larger the iterations of the communication, the better the performance. The above results demonstrate that our communication-based DKRR-RF is effective.

## 6 CONCLUSION

In this paper, we study the generalization properties of the combination of distributed learning and random features for ridge regression. We first improve the existing results of divide-and-conquer KRR with random features in expectation. Then, beyond the expectation, we derive generalization error bound in probability. Finally, we propose a novel effective communication strategy to further improve the learning performance of the combination method, and demonstrate the power of communications via both theoretical assessments and numerical experiments. Our results may open several venues for both theoretical and empirical work: (a) combine the approach with gradient algorithms such as preconditioned conjugate gradient Avron et al. (2017a) and multi-pass SGD (Carratino et al., 2018; Lin & Cevher, 2018; 2020); (b) replace synchronous distributed methods with asynchronous ones (Suresh et al., 2017); (c) consider the loss functions other than quadratic loss (Li et al., 2019e).

**Acknowledgment**
This work was supported in part by the National Natural Science Foundation of China NO. 62076234 and the Beijing Outstanding Young Scientist Program NO. BJJWZYJH012019100020098.

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

## A    APPENDIX: FAST LEARNING RATES

In the section, we will show that the fast learning rates can be achieved under favorable conditions.

Let

$$L_{\rho_{\mathcal{X}}}^2 = \left\{ f : \mathcal{X} \to \mathbb{R} \Big| \int_{\mathcal{X}} f^2(\mathbf{x}) d\rho_{\mathcal{X}} \le \infty \right\}$$

be the square integrable space, $\| \cdot \|_\rho^2$ be its norm. Denote the integral operator (Smale & Zhou, 2007) $L_K$ by

$$L_K f = \int_{\mathcal{X}} K(\mathbf{x}, \cdot) f(\mathbf{x}) d\rho_{\mathcal{X}}, \forall f \in L_{\rho_{\mathcal{X}}}^2.$$

**Assumption 1.** *For $\lambda > 0$, $\mathcal{N}(\lambda)$ is the effective dimension of the integral operator $L_K$ defined as $\mathcal{N}(\lambda) = \mathrm{Tr} \left( (L_K + \lambda I)^{-1} L_K \right)$, where $\mathrm{Tr}$ is the trace. Assuming there exists a constant $c \ge 1$, such that*

$$\mathcal{N}(\lambda) \le c\lambda^{-\gamma}, \gamma \in [0, 1]. \tag{7}$$

The effective dimension is a common assumption within the framework of learning theory (Caponnetto & Vito, 2007; Smale & Zhou, 2005; Rudi & Rosasco, 2017), which is used to measure the complexity of the hypothesis space. It is always satisfied for $\gamma = 1$ and $c = \kappa$. Equation 7 can control the variance of the estimator and is equivalent to the classic entropy and covering number conditions. In particular, it holds if the eigenvalues of integral operator $L_K$ decay as $i^{-1/\gamma}$, which is satisfied by the popular Gaussian and polynomial kernel functions. More details can be seen in (Steinwart & Christmann, 2008; Caponnetto & Vito, 2007; Rudi & Rosasco, 2017).

**Assumption 2.** *Let $f_\rho(\mathbf{x}) = \int_{\mathcal{Y}} y d\rho(y|\mathbf{x})$ be the regression function, $\rho(y|\mathbf{x})$ be the conditional distribution at $\mathbf{x}$ induced by $\rho$. For $\frac{1}{2} \le r \le 1$, assume there exists a $g \in L_{\rho_{\mathcal{X}}}^2$ such that*

$$f_\rho(\mathbf{x}) = L_K^r g(\mathbf{x}), \tag{8}$$

*where $L_K^r$ is the rth power of $L_K$.*

Regression function $f_\rho$ is the best function in $L_{\rho_{\mathcal{X}}}^2$, which is the primary objective in regression problem. Assumption 2 is used to measure the complexity of the regression function $f_\rho$, which is commonly used in approximation theory (Caponnetto & Vito, 2007). Equation 8 can be used to control the bias of the estimator, it requires the expansion of the regression function $f_\rho$ having coefficients that decay faster than the eigenvalues of integral operator $L_K$. The larger the value of $r$, the faster the coefficients decay. The case $r = 1/2$ means that $f_\rho \in \mathcal{H}_K$. More detail can be seen in (Rudi & Rosasco, 2017; Smale & Zhou, 2007).

**Theorem 4.** *Suppose $\psi$ is continuous, such that $|\psi(\mathbf{x}, \boldsymbol{\omega})| \le \tau$ almost surely, $\tau \in [1, \infty)$ and $|y| \le \zeta$. Under Assumptions 1-2 with $r \in [1/2, 1]$, $\gamma \in [0, 1]$, if $\lambda = \Omega(|\mathcal{D}|^{-\frac{1}{2r+\gamma}})$, $|\mathcal{D}_1| = \dots |\mathcal{D}_m|$,*

$$m \lesssim |\mathcal{D}|^{\frac{2r+\gamma-1}{2r+\gamma}} \text{ and } M \gtrsim |\mathcal{D}|^{\frac{1+\gamma(2r-1)}{2r+\gamma}}, \tag{9}$$

*then, for every $\delta \in (0, 1]$, with probability at least $1 - \delta$, we have*

$$\mathbb{E} \left[ \|\bar{f}_{M,\mathcal{D},\lambda}^0 - f_\rho\|_\rho^2 \right] = \mathcal{O} \left( |\mathcal{D}|^{-\frac{2r}{2r+\gamma}} \log^2 \frac{1}{\delta} \right).$$

The bound above is the same as the original KRR estimator and is optimal in a minimax sense (Caponnetto & Vito, 2007; Lin et al., 2017; Chang et al., 2017b). In the best case, when $r = 1$ and $\gamma = 0$, the rate $\mathcal{O}(1/|\mathcal{D}|)$ can be achieved by $\Omega(\sqrt{|\mathcal{D}|})$ random features and $\Omega(\sqrt{|\mathcal{D}|})$ partitions. In the worst case, that is $r = 1/2$ and $\gamma = 1$, which has been covered in Theorem 1.

**Theorem 5.** *Suppose $\psi$ is continuous, such that $|\psi(\mathbf{x}, \boldsymbol{\omega})| \le \tau$ almost surely, $\tau \in [1, \infty)$ and $|y| \le \zeta$. Under Assumptions 1-2 with $r \in [1/2, 1]$, $\gamma \in [0, 1]$, if $\lambda = \Omega(|\mathcal{D}|^{-\frac{1}{2r+\gamma}})$, $|\mathcal{D}_1| = \dots |\mathcal{D}_m|$,*

$$m \lesssim |\mathcal{D}|^{\frac{2r+\gamma-1}{4r+2\gamma}} \text{ and } M \gtrsim |\mathcal{D}|^{\frac{1+\gamma(2r-1)}{2r+\gamma}}, \tag{10}$$

*then, for every $\delta \in (0, 1]$, with probability at least $1 - \delta$, we have*

$$\|\bar{f}_{M,\mathcal{D},\lambda}^0 - f_\rho\|_\rho^2 = \mathcal{O} \left( |\mathcal{D}|^{-\frac{2r}{2r+\gamma}} \log^2 \frac{1}{\delta} \right).$$

One can see that the upper bound of $m$ is $|\mathcal{D}|^{\frac{2r+\gamma-1}{4r+2\gamma}}$, which is stricter than that of Theorem 4. In the best case, when $r = 1$ and $\gamma = 0$, the rate $\mathcal{O}(1/|\mathcal{D}|)$ can be achieved considering $\Omega(\sqrt{|\mathcal{D}|})$ random features and $\Omega(|\mathcal{D}|^{1/4})$ partitions. For the basic setting, that is $r = 1/2$ and $\gamma = 1$, which has been given in Theorem 2.

**Theorem 6.** *Suppose $\psi$ is continuous, such that $|\psi(\mathbf{x}, \boldsymbol{\omega})| \leq \tau$ almost surely, $\tau \in [1, \infty)$ and $|y| \leq \zeta$. Under Assumptions 1-2 with $r \in [1/2, 1]$, $\gamma \in [0, 1]$, if $\lambda = \Omega(|\mathcal{D}|^{-\frac{1}{2r+\gamma}})$, $|\mathcal{D}_1| = \dots |\mathcal{D}_m|$,*

$$m \lesssim |\mathcal{D}|^{\frac{(2r+\gamma-1)(p+1)}{(2r+\gamma)(p+2)}} \text{ and } M \gtrsim |\mathcal{D}|^{\frac{1+\gamma(2r-1)}{2r+\gamma}} \log \frac{1}{\delta}, \tag{11}$$

*then, for any $\delta \in (0, 1]$, with probability at least $1 - \delta$,*

$$\|\bar{f}_{M,\mathcal{D},\lambda}^p - f_\rho\|_\rho^2 = \mathcal{O}\left(|\mathcal{D}|^{-\frac{2r}{2r+\gamma}} \log^{p+2} \frac{1}{\delta}\right).$$

One can see that the communication can relax the restriction on the number of partitions. As $p \to \infty$, the partitions can reach $\Omega(|\mathcal{D}|^{\frac{2r+\gamma-1}{2r+\gamma}})$, which is the same as Theorem 4.

**Remark 5.** *In this paper, we focus on enlarging the number of local machines. In (Rudi & Rosasco, 2017; Rudi et al., 2018), they have proved that when generating random features in a data-dependent manner, fewer random features are required to obtain optimal learning. Thus, if we adopt the data-dependent manner to generate random features, we can further improve the performance of DKRR-RF-CM with fewer random features.*

# B    APPENDIX: NOTATION AND PRELIMINARY

In this paper we denote the operator norm by $\| \cdot \|$, and the square integrable norm by $\| \cdot \|_\rho$.

**Definition 1.**

$$S_M : \mathbb{R}^M \to L_{\rho_{\mathcal{X}}}^2, (S_M \mathbf{w})(\mathbf{x}) = \langle \mathbf{w}, \boldsymbol{\phi}_M(\mathbf{x}) \rangle,$$

$$S_M^* : L_{\rho_{\mathcal{X}}}^2 \to \mathbb{R}^M, S_M^* g = \int_{\mathcal{X}} \boldsymbol{\phi}_M(\mathbf{x}) g(\mathbf{x}) d\rho_{\mathcal{X}}(\mathbf{x}),$$

$$S_{M,\mathcal{D}}^* : L_{\rho_{\mathcal{X}}}^2 \to \mathbb{R}^M, S_{M,\mathcal{D}}^* g = \frac{1}{|\mathcal{D}|} \sum_{\mathbf{x}_j \in \mathcal{D}_{\mathcal{X}}} \boldsymbol{\phi}_M(\mathbf{x}_j) g(\mathbf{x}_j),$$

$$C_M : \mathbb{R}^M \to \mathbb{R}^M, C_M = \int_{\mathcal{X}} \boldsymbol{\phi}_M(\mathbf{x}) \boldsymbol{\phi}_M(\mathbf{x})^{\mathrm{T}} d\rho_{\mathcal{X}}(\mathbf{x}),$$

$$C_{M,\mathcal{D}} : \mathbb{R}^M \to \mathbb{R}^M, C_{M,\mathcal{D}} = \frac{1}{|\mathcal{D}|} \sum_{\mathbf{x}_j \in \mathcal{D}_{\mathcal{X}}} \boldsymbol{\phi}_M(\mathbf{x}_j) \boldsymbol{\phi}_M(\mathbf{x}_j)^{\mathrm{T}}.$$

**Lemma 1.** *$C_M$ and $C_{M,\mathcal{D}}$ are self-adjoint and positive operators, with spectrum is $[0, \tau^2]$. Moreover we have $C_M = S_M^* S_M$ and $C_{M,\mathcal{D}} = \boldsymbol{\Phi}_{M,\mathcal{D}} \boldsymbol{\Phi}_{M,\mathcal{D}}^{\mathrm{T}} = S_{M,\mathcal{D}}^* S_M$.*

*Proof.* $C_M$ and $C_{M,\mathcal{D}}$ are self-adjoint and positive operators, with spectrum is $[0, \tau^2]$, and

$$C_M = S_M^* S_M, C_{M,\mathcal{D}} = \boldsymbol{\Phi}_{M,\mathcal{D}} \boldsymbol{\Phi}_{M,\mathcal{D}}^{\mathrm{T}}$$

can be directly obtained from Caponnetto & Vito (2007); Smale & Zhou (2005; 2007); Rosasco et al. (2010); Rudi & Rosasco (2017); Lin et al. (2020).

In the following, we prove that $C_{M,\mathcal{D}} = S_{M,\mathcal{D}}^* S_M$. From the definitions of $S_M, S_{M,\mathcal{D}}^*$ and $C_{M,\mathcal{D}}$ in Definition 1, we have

$$\forall \boldsymbol{\beta} \in \mathbb{R}^M, (S_M \boldsymbol{\beta})(\cdot) = \langle \boldsymbol{\beta}, \boldsymbol{\phi}_M(\cdot) \rangle = \boldsymbol{\phi}_M(\cdot)^{\mathrm{T}} \boldsymbol{\beta},$$

and thus we can obtain that,

$$S_{M,\mathcal{D}}^* S_M \boldsymbol{\beta} = \frac{1}{|\mathcal{D}|} \sum_{\mathbf{x}_j \in \mathcal{D}_{\mathcal{X}}} \boldsymbol{\phi}_M(\mathbf{x}_j) \boldsymbol{\phi}_M(\mathbf{x}_j)^{\mathrm{T}} \boldsymbol{\beta} = C_{M,\mathcal{D}} \boldsymbol{\beta}.$$

So, the Equation $C_{M,\mathcal{D}} = S_{M,\mathcal{D}}^* S_M$ holds. □

**Definition 2.**

$$f_{M,\mathcal{D},\lambda} = \mathbf{w}_{M,\mathcal{D},\lambda}^{\mathrm{T}}\phi_M(\cdot), \mathbf{w}_{M,\mathcal{D},\lambda} = \underset{\mathbf{w}\in\mathbb{R}^M}{\arg\min}\left\{\frac{1}{|\mathcal{D}|}\sum_{z_i\in\mathcal{D}}\left(\mathbf{w}^{\mathrm{T}}\phi_M(\mathbf{x}_i) - y_i\right)^2 + \lambda\|\mathbf{w}\|^2\right\};$$

$$f_{M,\mathcal{D},\lambda}^{\diamond} = \mathbf{w}_{M,\mathcal{D},\lambda}^{\diamond\mathrm{T}}\phi_M(\cdot), \mathbf{w}_{M,\mathcal{D},\lambda}^{\diamond} = \underset{\mathbf{w}\in\mathbb{R}^M}{\arg\min}\left\{\frac{1}{|\mathcal{D}|}\sum_{z_i\in\mathcal{D}}\left(\mathbf{w}^{\mathrm{T}}\phi_M(\mathbf{x}_i) - f_\rho(\mathbf{x}_i)\right)^2 + \lambda\|\mathbf{w}\|^2\right\};$$

$$f_{M,\lambda} = \mathbf{w}_{M,\lambda}^{\mathrm{T}}\phi_M(\cdot), \mathbf{w}_{M,\lambda} = \underset{\mathbf{w}\in\mathbb{R}^M}{\arg\min}\left\{\int_{\mathcal{X}}\left(\mathbf{w}^{\mathrm{T}}\phi_M(\mathbf{x}) - f_\rho(\mathbf{x})\right)^2 d\rho_{\mathcal{X}}(\mathbf{x}) + \lambda\|\mathbf{w}\|^2\right\};$$

$$f_\lambda = \underset{f\in\mathcal{H}_K}{\arg\min}\left\{\int_{\mathcal{X}}\left(f(\mathbf{x}) - f_\rho(\mathbf{x})\right)^2 d\rho_{\mathcal{X}}(\mathbf{x}) + \lambda\|f\|_K^2\right\}. \tag{12}$$

**Remark 6** (From Caponnetto & Vito (2007); Smale & Zhou (2007); Li et al. (2019b); Lin et al. (2020)).

$$f_{M,\mathcal{D},\lambda} = S_M\mathbf{w}_{M,\mathcal{D},\lambda}, \mathbf{w}_{M,\mathcal{D},\lambda} = (C_{M,\mathcal{D}} + \lambda I)^{-1}\boldsymbol{\Phi}_{M,\mathcal{D}}\bar{\mathbf{y}}_{\mathcal{D}};$$
$$f_{M,\mathcal{D},\lambda}^{\diamond} = S_M\mathbf{w}_{M,\mathcal{D},\lambda}^{\diamond}, \mathbf{w}_{M,\mathcal{D},\lambda}^{\diamond} = (C_{M,\mathcal{D}} + \lambda I)^{-1}S_{M,\mathcal{D}}^* f_\rho; \tag{13}$$
$$f_{M,\lambda} = S_M\mathbf{w}_{M,\lambda}, \mathbf{w}_{M,\lambda} = (C_M + \lambda I)^{-1}S_M^* f_\rho.$$

**Definition 3.** *The maximum random feature dimension is denoted as*

$$\mathcal{N}_\infty(\lambda) = \sup_{\boldsymbol{\omega}\in\Omega}\|(L_K + \lambda I)^{-1/2}\psi(\cdot,\boldsymbol{\omega})\|_\rho^2, \lambda > 0.$$

**Remark 7** (From Rudi & Rosasco (2017)). $\mathcal{N}_\infty(\lambda) \le \tau^2\lambda^{-1}$ *is always satisfied for every $\lambda > 0$.*

## C APPENDIX: PROOF OF THEOREM 4

### C.1 APPENDIX: ERROR DECOMPOSITION FOR DKRR-RF IN EXPECTATION

**Proposition 1.** *Let $\bar{f}_{M,\mathcal{D},\lambda}^0$, and $f_{M,\mathcal{D},\lambda}^{\diamond}$, $f_{M,\lambda}$, $f_\lambda$ be defined by 3 and 12, respectively. Then, we have*

$$\mathbb{E}\left[\|\bar{f}_{M,\mathcal{D},\lambda}^0 - f_\rho\|_\rho^2\right] \le 3\left[\|f_{M,\lambda} - f_\lambda\|_\rho^2\right] + 3\left[\|f_\lambda - f_\rho\|_\rho^2\right]$$
$$+ 3\sum_{j=1}^m\frac{|\mathcal{D}_j|^2}{|\mathcal{D}|^2}\mathbb{E}\left[\|f_{M,\mathcal{D}_j,\lambda} - f_{M,\lambda}\|_\rho^2\right] + 3\sum_{j=1}^m\frac{|\mathcal{D}_j|}{|\mathcal{D}|}\mathbb{E}\left[\|f_{M,\mathcal{D}_j,\lambda}^{\diamond} - f_{M,\lambda}\|_\rho^2\right].$$

*Proof.* Let $\bar{f}_{M,\mathcal{D},\lambda}^0$, and $f_{M,\mathcal{D},\lambda}^{\diamond}$, $f_{M,\lambda}$ be defined by 3 and 12, respectively. According to the Proposition 5 of (Chang et al., 2017b) or Lemma 20 of (Li et al., 2019b), we have

$$\mathbb{E}\left[\|\bar{f}_{M,\mathcal{D},\lambda}^0 - f_{M,\lambda}\|_\rho^2\right]$$
$$\le \sum_{j=1}^m\frac{|\mathcal{D}_j|^2}{|\mathcal{D}|^2}\mathbb{E}\left[\|f_{M,\mathcal{D}_j,\lambda} - f_{M,\lambda}\|_\rho^2\right] + \sum_{j=1}^m\frac{|\mathcal{D}_j|}{|\mathcal{D}|}\mathbb{E}\left[\|f_{M,\mathcal{D},\lambda}^{\diamond} - f_{M,\lambda}\|_\rho^2\right]. \tag{14}$$

Note that $(a + b + c)^2 \le 3a^2 + 3b^2 + 3c^2, \forall a, b, c \ge 0$. Thus, we have

$$\mathbb{E}\left[\|\bar{f}_{M,\mathcal{D},\lambda}^0 - f_\rho\|_\rho^2\right] = \mathbb{E}\left[\|\bar{f}_{M,\mathcal{D},\lambda}^0 - f_{M,\lambda} + f_{M,\lambda} - f_\lambda + f_\lambda - f_\rho\|_\rho^2\right]$$
$$\le 3\mathbb{E}\left[\|\bar{f}_{M,\mathcal{D},\lambda}^0 - f_{M,\lambda}\|_\rho^2\right] + 3\left[\|f_{M,\lambda} - f_\lambda\|_\rho^2\right] + 3\left[\|f_\lambda - f_\rho\|_\rho^2\right].$$

Combining the above inequality and Eq. 14, we can prove this proposition. $\square$

**Proposition 2.** *The follows hold:*

$$\sqrt{\lambda}\|\mathbf{w}_{M,\mathcal{D},\lambda} - \mathbf{w}_{M,\mathcal{D},\lambda}^{\diamond}\| \le \mathcal{J}_{M,\mathcal{D}}(\mathcal{R}_{M,\mathcal{D}} + \mathcal{K}_{M,\mathcal{D}})$$

*and*

$$\|f_{M,\mathcal{D},\lambda} - f_{M,\mathcal{D},\lambda}^{\diamond}\|_\rho \le \mathcal{J}_{M,\mathcal{D}}^2(\mathcal{R}_{M,\mathcal{D}} + \mathcal{K}_{M,\mathcal{D}}).$$

*where $\mathcal{R}_{M,\mathcal{D}} := \|(C_M + \lambda I)^{-1/2}(\boldsymbol{\Phi}_{M,\mathcal{D}}\bar{\mathbf{y}}_{\mathcal{D}} - S_M^* f_\rho)\|$, $\mathcal{J}_{M,\mathcal{D}} := \|(C_{M,\mathcal{D}} + \lambda I)^{-1/2}(C_M + \lambda I)^{1/2}\|$, $\mathcal{K}_{M,\mathcal{D}} := \|(C_M + \lambda I)^{-1/2}(S_M^* f_\rho - S_{M,\mathcal{D}}^* f_\rho)\|$.*

*Proof.* From 13, we know that $\mathbf{w}_{M,\mathcal{D},\lambda} = (C_{M,\mathcal{D}} + \lambda I)^{-1}\mathbf{\Phi}_{M,\mathcal{D}}\bar{\mathbf{y}}_{\mathcal{D}}$ and $\mathbf{w}^{\diamond}_{M,\mathcal{D},\lambda} = (C_{M,\mathcal{D}} + \lambda I)^{-1}S^*_{M,\mathcal{D}}f_\rho$, so we have

$$\mathbf{w}_{M,\mathcal{D},\lambda} - \mathbf{w}^{\diamond}_{M,\mathcal{D},\lambda} = (C_{M,\mathcal{D}} + \lambda I)^{-1}(\mathbf{\Phi}_{M,\mathcal{D}}\bar{\mathbf{y}}_{\mathcal{D}} - S^*_{M,\mathcal{D}}f_\rho)$$
$$= (C_{M,\mathcal{D}} + \lambda I)^{-1/2}(C_{M,\mathcal{D}} + \lambda I)^{-1/2}(C_M + \lambda I)^{1/2}(C_M + \lambda I)^{-1/2}(\mathbf{\Phi}_{M,\mathcal{D}}\bar{\mathbf{y}}_{\mathcal{D}} - S^*_{M,\mathcal{D}}f_\rho).$$
(15)

Note $(C_{M,\mathcal{D}} + \lambda I)^{-1/2}$ is a self-adjoint and positive operator, so $\|(C_{M,\mathcal{D}} + \lambda I)^{-1/2}\| \le 1/\sqrt{\lambda}$, thus, we can obtain that

$$\|\mathbf{w}_{M,\mathcal{D},\lambda} - \mathbf{w}^{\diamond}_{M,\mathcal{D},\lambda}\| \le \frac{1}{\sqrt{\lambda}}\mathcal{J}_{M,\mathcal{D}}\|(C_M + \lambda I)^{-1/2}(\mathbf{\Phi}_{M,\mathcal{D}}\bar{\mathbf{y}}_{\mathcal{D}} - S^*_{M,\mathcal{D}}f_\rho)\|$$

$$= \frac{1}{\sqrt{\lambda}}\mathcal{J}_{M,\mathcal{D}}\|(C_M + \lambda I)^{-1/2}(\mathbf{\Phi}_{M,\mathcal{D}}\bar{\mathbf{y}}_{\mathcal{D}} - S^*_{M,\mathcal{D}}f_\rho + S^*_{M,\mathcal{D}}f_\rho - S^*_{M,\mathcal{D}}f_\rho)\|$$ (16)

$$\le \frac{1}{\sqrt{\lambda}}\mathcal{J}_{M,\mathcal{D}}\left(\mathcal{R}_{M,\mathcal{D}} + \mathcal{K}_{M,\mathcal{D}}\right).$$

Note that $f_{M,\mathcal{D},\lambda} - f^{\diamond}_{M,\mathcal{D},\lambda} = S_M(\mathbf{w}_{M,\mathcal{D},\lambda} - \mathbf{w}^{\diamond}_{M,\mathcal{D},\lambda})$, by 15, we have

$$f_{M,\mathcal{D},\lambda} - f^{\diamond}_{M,\mathcal{D},\lambda} = S_M(\mathbf{w}_{M,\mathcal{D},\lambda} - \mathbf{w}^{\diamond}_{M,\mathcal{D},\lambda})$$
$$= S_M(C_M + \lambda I)^{-1/2}(C_M + \lambda I)^{1/2}(C_{M,\mathcal{D}} + \lambda I)^{-1/2}(C_{M,\mathcal{D}} + \lambda I)^{-1/2}(C_M + \lambda I)^{1/2}$$ (17)
$$(C_M + \lambda I)^{-1/2}(\mathbf{\Phi}_{M,\mathcal{D}}\bar{\mathbf{y}}_{\mathcal{D}} - S^*_{M,\mathcal{D}}f_\rho + S^*_{M,\mathcal{D}}f_\rho - S^*_{M,\mathcal{D}}f_\rho).$$

Note that

$$\|S_M(C_M + \lambda I)^{-1/2}\| = \|(C_M + \lambda I)^{-1/2}S^*_M S_M(C_M + \lambda I)^{-1/2}\|^{1/2}$$
$$= \|(C_M + \lambda I)^{-1/2}C_M(C_M + \lambda I)^{-1/2}\|^{1/2} \le 1.$$

So, by Eq 17, we have

$$\|f_{M,\mathcal{D},\lambda} - f^{\diamond}_{M,\mathcal{D},\lambda}\|_\rho \le \mathcal{J}^2_{M,\mathcal{D}}\left(\mathcal{R}_{M,\mathcal{D}} + \mathcal{K}_{M,\mathcal{D}}\right).$$

$\square$

**Proposition 3.** *The follows hold:*
$$\sqrt{\lambda}\|\mathbf{w}^{\diamond}_{M,\mathcal{D},\lambda} - \mathbf{w}_{M,\lambda}\| \le \|f_{M,\lambda} - f_\rho\|_\rho + \mathcal{J}_{M,\mathcal{D}}\|f_{M,\lambda} - f_\rho\|_\rho$$
*and*
$$\|f^{\diamond}_{M,\mathcal{D},\lambda} - f_{M,\lambda}\|_\rho \le (\mathcal{J}_{M,\mathcal{D}} + \mathcal{J}^2_{M,\mathcal{D}})\|f_{M,\lambda} - f_\rho\|_\rho,$$
*where $\mathcal{J}_{M,\mathcal{D}} := \|(C_{M,\mathcal{D}} + \lambda I)^{-1/2}(C_M + \lambda I)^{1/2}\|$.*

*Proof.* By Remark 6, we have
$$\mathbf{w}^{\diamond}_{M,\mathcal{D},\lambda} - \mathbf{w}_{M,\lambda} = (C_{M,\mathcal{D}} + \lambda I)^{-1}S^*_{M,\mathcal{D}}f_\rho - (C_M + \lambda I)^{-1}S^*_M f_\rho$$
$$= (C_{M,\mathcal{D}} + \lambda I)^{-1}[S^*_{M,\mathcal{D}}f_\rho - S^*_M f_\rho] + [(C_{M,\mathcal{D}} + \lambda I)^{-1} - (C_M + \lambda I)^{-1}]S^*_M f_\rho.$$
Note that for any self-adjoint and positive operators $A$ and $B$,
$$A^{-1} - B^{-1} = A^{-1}(B - A)B^{-1}, A^{-1} - B^{-1} = B^{-1}(B - A)A^{-1},$$ (18)
so we have
$$\mathbf{w}^{\diamond}_{M,\mathcal{D},\lambda} - \mathbf{w}_{M,\lambda}$$
$$= (C_{M,\mathcal{D}} + \lambda I)^{-1}[S^*_{M,\mathcal{D}}f_\rho - S^*_M f_\rho] + (C_{M,\mathcal{D}} + \lambda I)^{-1}(C_M - C_{M,\mathcal{D}})\mathbf{w}_{M,\lambda}$$
From Lemma 1, we know that $C_M = S^*_M S_M$ and $C_{M,\mathcal{D}} = \mathbf{\Phi}_{M,\mathcal{D}}\mathbf{\Phi}^{\mathrm{T}}_{M,\mathcal{D}} = S^*_{M,\mathcal{D}}S_M$, thus we can obtain that
$$\mathbf{w}^{\diamond}_{M,\mathcal{D},\lambda} - \mathbf{w}_{M,\lambda}$$
$$= (C_{M,\mathcal{D}} + \lambda I)^{-1}[S^*_{M,\mathcal{D}}f_\rho - S^*_M f_\rho] + (C_{M,\mathcal{D}} + \lambda I)^{-1}(S^*_M S_M \mathbf{w}_{M,\lambda} - S^*_{M,\mathcal{D}}S_M \mathbf{w}_{M,\lambda})$$
$$= (C_{M,\mathcal{D}} + \lambda I)^{-1}[S^*_{M,\mathcal{D}}f_\rho - S^*_{M,\mathcal{D}}S_M \mathbf{w}_{M,\lambda}] + (C_{M,\mathcal{D}} + \lambda I)^{-1}[S^*_M S_M \mathbf{w}_{M,\lambda} - S^*_M f_\rho]$$ (19)
$$= (C_{M,\mathcal{D}} + \lambda I)^{-1}[S^*_{M,\mathcal{D}}f_\rho - S^*_{M,\mathcal{D}}f_{M,\lambda}] + (C_{M,\mathcal{D}} + \lambda I)^{-1}[S^*_M f_{M,\lambda} - S^*_M f_\rho]$$
$$= (C_{M,\mathcal{D}} + \lambda I)^{-1}S^*_{M,\mathcal{D}}[f_\rho - f_{M,\lambda}] + (C_{M,\mathcal{D}} + \lambda I)^{-1}S^*_M[f_{M,\lambda} - f_\rho].$$

Thus, we have

$$\|\mathbf{w}_{M,\mathcal{D},\lambda}^{\diamond} - \mathbf{w}_{M,\lambda}\| \leq \left( \|(C_{M,\mathcal{D}} + \lambda I)^{-1} S_{M,\mathcal{D}}^*\| + \|(C_{M,\mathcal{D}} + \lambda I)^{-1} S_M^*\| \right) \|f_{M,\lambda} - f_\rho\|_\rho. \quad (20)$$

Note that

$$\|(C_{M,\mathcal{D}} + \lambda I)^{-1/2} S_{M,\mathcal{D}}^*\| \leq \|(C_{M,\mathcal{D}} + \lambda I)^{-1/2} C_{M,\mathcal{D}} (C_{M,\mathcal{D}} + \lambda I)^{-1/2}\|^{1/2} \leq 1$$

and

$$\begin{aligned}
\|(C_{M,\mathcal{D}} + \lambda I)^{-1/2} S_M^*\| =& \|(C_{M,\mathcal{D}} + \lambda I)^{-1/2} (C_M + \lambda I)^{1/2} (C_M + \lambda I)^{-1/2} S_M^*\| \\
&\leq \mathcal{J}_{M,\mathcal{D}} \|(C_M + \lambda I)^{-1/2} S_M^*\| \leq \mathcal{J}_{M,\mathcal{D}},
\end{aligned}$$

since $\|(C_M + \lambda I)^{-1/2} S_M^*\| = \|(C_M + \lambda I)^{-1/2} C_M (C_M + \lambda I)^{-1/2}\|^{1/2} \leq 1$. Substituting the above two inequalities into Eq. 20, we have

$$\|\mathbf{w}_{M,\mathcal{D},\lambda}^{\diamond} - \mathbf{w}_{M,\lambda}\| \leq \frac{1}{\sqrt{\lambda}} (1 + \mathcal{J}_{M,\mathcal{D}}) \|f_{M,\lambda} - f_\rho\|_\rho,$$

which prove the first result of Proposition 3.

In the following, we will prove the second result. By Eq. 19, we have

$$\begin{aligned}
f_{M,\mathcal{D},\lambda}^{\diamond} - f_{M,\lambda} =& S_M (\mathbf{w}_{M,\mathcal{D},\lambda}^{\diamond} - \mathbf{w}_{M,\lambda}) \\
=& S_M (C_{M,\mathcal{D}} + \lambda I)^{-1} S_{M,\mathcal{D}}^* [f_\rho - f_{M,\lambda}] + S_M (C_{M,\mathcal{D}} + \lambda I)^{-1} S_M^* [f_{M,\lambda} - f_\rho] \\
=& S_M (C_M + \lambda I)^{-1/2} (C_M + \lambda I)^{1/2} (C_{M,\mathcal{D}} + \lambda I)^{-1/2} (C_{M,\mathcal{D}} + \lambda I)^{-1/2} S_{M,\mathcal{D}}^* [f_\rho - f_{M,\lambda}] \\
&+ S_M (C_M + \lambda I)^{-1/2} (C_M + \lambda I)^{1/2} (C_{M,\mathcal{D}} + \lambda I)^{-1/2} (C_{M,\mathcal{D}} + \lambda I)^{-1/2} (C_M + \lambda I)^{1/2} \\
&\quad (C_M + \lambda I)^{-1/2} S_M^* [f_{M,\lambda} - f_\rho]
\end{aligned}$$

Note that $\|S_M (C_M + \lambda I)^{-1/2}\| = \|(C_M + \lambda I)^{-1/2} C_M (C_M + \lambda I)^{-1/2}\|^{1/2} \leq 1$, $\|(C_{M,\mathcal{D}} + \lambda I)^{-1/2} S_{M,\mathcal{D}}^*\| = \|(C_{M,\mathcal{D}} + \lambda I)^{-1/2} C_{M,\mathcal{D}} (C_{M,\mathcal{D}} + \lambda I)^{-1/2}\|^{1/2} \leq 1$, and $\|(C_M + \lambda I)^{-1/2} S_M^*\| = \|(C_M + \lambda I)^{-1/2} C_M (C_M + \lambda I)^{-1/2}\|^{1/2} \leq 1$, so we have

$$\|f_{M,\mathcal{D},\lambda}^{\diamond} - f_{M,\lambda}\|_\rho \leq (\mathcal{J}_{M,\mathcal{D}} + \mathcal{J}_{M,\mathcal{D}}^2) \|f_{M,\lambda} - f_\rho\|_\rho. \quad (21)$$

$\square$

**Proposition 4.** *The follows hold:*

$$\sqrt{\lambda} \|\mathbf{w}_{M,\mathcal{D},\lambda} - \mathbf{w}_{M,\lambda}\| \leq \mathcal{J}_{M,\mathcal{D}} (\mathcal{R}_{M,\mathcal{D}} + \mathcal{K}_{M,\mathcal{D}}) + (1 + \mathcal{J}_{M,\mathcal{D}}) \|f_{M,\lambda} - f_\rho\|_\rho,$$

*and*

$$\|f_{M,\mathcal{D},\lambda} - f_{M,\lambda}\|_\rho \leq \mathcal{J}_{M,\mathcal{D}}^2 (\mathcal{R}_{M,\mathcal{D}} + \mathcal{K}_{M,\mathcal{D}}) + (\mathcal{J}_{M,\mathcal{D}} + \mathcal{J}_{M,\mathcal{D}}^2) \|f_{M,\lambda} - f_\rho\|_\rho.$$

*where* $\mathcal{R}_{M,\mathcal{D}} := \|(C_M + \lambda I)^{-1/2} (\mathbf{\Phi}_{M,\mathcal{D}} \bar{\mathbf{y}}_{\mathcal{D}} - S_M^* f_\rho)\|$, $\mathcal{J}_{M,\mathcal{D}} := \|(C_{M,\mathcal{D}} + \lambda I)^{-1/2} (C_M + \lambda I)^{1/2}\|$, $\mathcal{K}_{M,\mathcal{D}} := \|(C_M + \lambda I)^{-1/2} (S_M^* f_\rho - S_{M,\mathcal{D}}^* f_\rho)\|$.

*Proof.* Note that

$$\|\mathbf{w}_{M,\mathcal{D},\lambda} - \mathbf{w}_{M,\lambda}\| \leq \|\mathbf{w}_{M,\mathcal{D},\lambda} - \mathbf{w}_{M,\mathcal{D},\lambda}^{\diamond}\| + \|\mathbf{w}_{M,\mathcal{D},\lambda}^{\diamond} - \mathbf{w}_{M,\lambda}\|$$

and

$$\|f_{M,\mathcal{D},\lambda} - f_{M,\lambda}\|_\rho \leq \|f_{M,\mathcal{D},\lambda} - f_{M,\mathcal{D},\lambda}^{\diamond}\|_\rho + \|f_{M,\mathcal{D},\lambda}^{\diamond} - f_{M,\lambda}\|_\rho.$$

Combining Propositions 2 and 3, we can prove this result. $\square$

## C.2 Proof of Theorem 4

**Lemma 2** (Lemma 23 in (Li et al., 2019b), can be also seen in (Rudi & Rosasco, 2017)). *For $\delta \in (0, 1]$ and $\lambda > 0$, when*

$$M \geq \Omega \left( \left( \frac{\mathcal{N}(\lambda)}{\lambda} \right)^{2r-1} \left( \mathcal{N}_\infty(\lambda) \log \frac{1}{\lambda} \right)^{2-2r} \vee (\mathcal{N}_\infty(\lambda)) \log \frac{1}{\lambda \delta} \right).$$

*Then, with probability at least $1 - \delta$, we have*

$$\|f_{M,\lambda} - f_\lambda\|_\rho^2 \leq c\lambda^{2r},$$

*where $c$ is a constant.*

**Lemma 3** (Theorem 4 in (Smale & Zhou, 2005)). *Under Assumption 2, for $r \in [1/2, 1]$, we have*

$$\|f_\lambda - f_\rho\|_\rho^2 \leq c\lambda^{2r},$$

*where $c$ is a constant.*

**Lemma 4** (Lemma 6 in (Rudi & Rosasco, 2017)). *For $\delta \in (0, 1]$, with probability at least $1 - \delta$, we have*

$$\mathcal{R}_{M,\mathcal{D}} := \left\| (C_M + \lambda I)^{-1/2} \left( \mathbf{\Phi}_{M,\mathcal{D}} \bar{\mathbf{y}}_\mathcal{D} - S_M^* f_\rho \right) \right\| = \mathcal{O} \left( \left( \frac{1}{\sqrt{\lambda}|\mathcal{D}|} + \sqrt{\frac{\mathcal{N}_M(\lambda)}{|\mathcal{D}|}} \right) \log \frac{1}{\delta} \right),$$

*where $\mathcal{N}_M(\lambda) := \mathrm{Tr}\left( (L_M + \lambda I)^{-1} L_M \right)$, $L_M$ is the integral operator associated with the approximate kernel function $K_M$, $(L_M f)(\mathbf{x}) = \int_\mathcal{X} K_M(\mathbf{x}, \mathbf{x}') f(\mathbf{x}') d\rho_\mathcal{X}(\mathbf{x}')$.*

**Lemma 5** (Proposition 10 in (Rudi & Rosasco, 2017)). *For any $\delta \in (0, 1]$, $M \geq \Omega\left( \mathcal{N}_\infty(\lambda) \log \frac{1}{\lambda \delta} \right)$, then with probability at least $1 - \delta$,*

$$|\mathcal{N}_M(\lambda) - \mathcal{N}(\lambda)| \leq 1.55 \mathcal{N}(\lambda), \tag{22}$$

*where $\mathcal{N}_M(\lambda) := \mathrm{Tr}\left( (L_M + \lambda I)^{-1} L_M \right)$.*

**Lemma 6** (Lemma E.2, (Blanchard & Krämer, 2010)). *For any self-adjoint and positive semidefinite operators $A$ and $B$, if there exists $\eta > 0$ such that the following inequality holds*

$$\|(A + \lambda I)^{-1/2}(B - A)(A + \lambda I)^{-1/2}\| \leq 1 - \eta,$$

*then*

$$\|(A + \lambda I)^{1/2}(B + \lambda I)^{-1/2}\| \leq \frac{1}{\sqrt{\eta}}.$$

**Proposition 5.** *For $\delta \in (0, 1]$, with probability at least $1 - \delta$, we have*

$$\mathcal{K}_{M,\mathcal{D}} := \|(C_M + \lambda I)^{-1/2}(S_M^* f_\rho - S_{M,\mathcal{D}}^* f_\rho)\| \leq \frac{2\tau \zeta \log \frac{1}{\delta}}{3|\mathcal{D}|\sqrt{\lambda}} + 2\zeta \sqrt{\frac{\mathcal{N}_M(\lambda)}{|\mathcal{D}|}},$$

*where $\mathcal{N}_M(\lambda) := \mathrm{Tr}\left( (L_M + \lambda I)^{-1} L_M \right)$.*

The proof closely follows the proof of Lemma 6 in (Rudi & Rosasco, 2017).

*Proof.* Define $\mu_i = (C_M + \lambda I)^{1/2} S_M^* f_\rho - (C_M + \lambda I)^{1/2} \phi_M(\mathbf{x}_i) f_\rho(\mathbf{x}_i)$. Note that

$$(C_M + \lambda I)^{1/2}(S_M^* f_\rho - S_{M,\mathcal{D}}^* f_\rho) = \frac{1}{|\mathcal{D}|} \sum_{i=1}^{|\mathcal{D}|} \mu_i.$$

Since $\mu_1, \ldots, \mu_{|\mathcal{D}|}$ are independent and identically distributed random vector, and

$$\mathbb{E}\mu_i = \int_\mathcal{X} (C_M + \lambda I)^{1/2} \phi_M(\mathbf{x}) f_\rho(\mathbf{x}) d\rho_\mathcal{X} - \int_\mathcal{X} (C_M + \lambda I)^{1/2} \phi_M(\mathbf{x}_i) f_\rho(\mathbf{x}_i) d\rho_\mathcal{X} = 0.$$

To apply the Bernstein inequality (Arcones, 1995; Rudi & Rosasco, 2017) for random vectors, we need to bound their moments. Note that

$$\|(C_M + \lambda I)^{1/2} \phi_M(\mathbf{x}) f_\rho(\mathbf{x})\| \leq \frac{\tau \zeta}{\sqrt{\lambda}},$$

and

$$\mathbb{E}\mu_i^2 \leq 2 \int_{\mathcal{X}} \|(C_M + \lambda I)^{1/2} \phi_M(\mathbf{x})\|^2 \|f_\rho(\mathbf{x})\|_\rho^2 d\rho_{\mathcal{X}}(\mathbf{x})$$

$$\leq 2\zeta^2 \int_{\mathcal{X}} \|(C_M + \lambda I)^{1/2} \phi_M(\mathbf{x})\|^2 d\rho_{\mathcal{X}}(\mathbf{x}) \leq 2\zeta^2 \mathcal{N}_M(\lambda).$$

Thus, using the Bernstein inequality (Arcones, 1995; Rudi & Rosasco, 2017), for any $\delta \in (0, 1]$, with $1 - \delta$, we have

$$\|(C_M + \lambda I)^{-1/2}(S_M^* f_\rho - S_{M,\mathcal{D}}^* f_\rho)\| \leq \frac{2\tau\zeta \log \frac{1}{\delta}}{3|\mathcal{D}|\sqrt{\lambda}} + 2\zeta \sqrt{\frac{\mathcal{N}_M(\lambda)}{|\mathcal{D}|}}.$$

$\square$

**Lemma 7** (Lemma 2 in (Smale & Zhou, 2007)). *Let $\mathcal{H}$ be a Hilbert space and $\xi$ be a random variable on $\mathcal{Z}, \rho$ with values in $\mathcal{H}$. Assume $\|\xi\| \leq M < \infty$ almost surely. Denote $\sigma(\xi) = \mathbb{E}(\xi^2)$. Let $\{z_i\}_{i=1}^n$ be independent random drawers of $\rho$. For any $0 < \delta < 1$, with confidence $1 - \delta$,*

$$\left\| \frac{1}{n} \sum_{i=1}^m [\xi - \mathbb{E}(\xi)] \right\| \leq \frac{2M \log(2/\delta)}{n} + \sqrt{\frac{2\|\sigma(\xi)\|^2 \log(2/\delta)}{n}}.$$

**Proposition 6.** *For any $\delta > 0$, with probability at least $1 - \delta$, we have*

$$\mathcal{Q}_{M,\mathcal{D}} := \|(C_M + \lambda I)^{-1/2}(C_M - C_{M,\mathcal{D}})(C_M + \lambda)^{-1/2}\|$$

$$\leq \frac{2 \log^2(2/\delta)(\mathcal{N}_\infty(\lambda) + 1)}{|\mathcal{D}|} + \sqrt{\frac{2 \log(2/\delta)(\mathcal{N}_\infty(\lambda) + 1)}{|\mathcal{D}|}}.$$

*and*

$$\mathcal{L}_{M,\mathcal{D}} := \|(C_M + \lambda I)^{-1}(C_M - C_{M,\mathcal{D}})\|$$

$$\leq \frac{2 \log^2(2/\delta)(\mathcal{N}_\infty(\lambda) + 1)}{|\mathcal{D}|} + \sqrt{\frac{2 \log(2/\delta)(\mathcal{N}_\infty(\lambda) + 1)}{|\mathcal{D}|}}.$$

*where $\mathcal{N}_\infty(\lambda) = \sup_{\boldsymbol{\omega} \in \Omega} \|(L_K + \lambda I)^{-1/2} \psi(\cdot, \boldsymbol{\omega})\|_\rho^2$, $c_1$ and $c_2$ are two constants.*

To prove Proposition 6, we first prove the following lemma (a similar technique can be found in (Hsu et al., 2012; Rudi et al., 2013; Caponnetto & Yao, 2006; Rudi & Rosasco, 2017)).

The similar result for the matrix case was first proved in (Hsu et al., 2012), and later was extended to the operator case in (Rudi et al., 2013; Rudi & Rosasco, 2017).

**Lemma 8.** *Let $\zeta_1, \ldots, \zeta_n$ with $n \geq 1$, be i.i.d random vectors on a separable Hilbert spaces $\mathcal{H}$ such that $H = \mathbb{E}\zeta \otimes \zeta$ is a trace class, and for any $\lambda$ there exists $\mathcal{N}_\infty(\lambda) < \infty$ such that $\langle \zeta, (H + \lambda I)^{-1}\zeta \rangle \leq \mathcal{N}_\infty(\lambda)$. Denote $H_n$ as $\frac{1}{n} \sum_{i=1}^n \zeta_i \otimes \zeta_i$. Then for any $\delta \geq 0$, with probability at least $1 - 2\delta$, the following holds*

$$\left\|(H + \lambda I)^{-1/2}(H - H_n)(H + \lambda)^{-1/2}\right\| \leq \frac{2 \log^2(2/\delta)(\mathcal{N}_\infty(\lambda) + 1)}{n} + \sqrt{\frac{2 \log(2/\delta)(\mathcal{N}_\infty(\lambda) + 1)}{n}}.$$

*Proof.* Let $H_\lambda = H + \lambda I$, $\eta = H_\lambda^{-1/2} H H_\lambda^{-1/2}$, $\xi_i = \eta - H_\lambda^{-1/2}\zeta_i \otimes H_\lambda^{-1/2}\zeta_i$. One can see that $\mathbb{E}\xi_i = 0$. Note that

$$\|\eta - H_\lambda^{-1/2}\zeta_i \otimes H_\lambda^{-1/2}\zeta_i\| \leq \|\eta\| + \langle \zeta_i, H_\lambda^{-1/2}\zeta_i \rangle \leq 1 + \mathcal{N}_\infty(\lambda),$$

and

$$
\begin{aligned}
\|\mathbb{E}[\xi_i^2]\| &= \left\| \mathbb{E}\left[ \langle \zeta_i, H_\lambda^{-1/2}\zeta_i \rangle H_\lambda^{-1/2}\zeta_i \otimes H_\lambda^{-1/2}\zeta_i \right] - H_\lambda^{-2}H^2 \right\| \\
&\leq \mathcal{N}_\infty(\lambda) \left\| \mathbb{E}[H_\lambda^{-1/2}\zeta_i \otimes H_\lambda^{-1/2}\zeta_i] \right\| + \left\| H_\lambda^{-2}H^2 \right\| \\
&\leq \mathcal{N}_\infty(\lambda)\|H_\lambda^{-1/2}H\| + \left\| H_\lambda^{-2}H^2 \right\| \\
&\leq \mathcal{N}_\infty(\lambda) + 1.
\end{aligned}
$$

Thus, substituting the above two inequalities to Lemma 7 (Lemma 2 in (Smale & Zhou, 2007)), which finishes the proof. □

*Proof of Proposition 6.* Since $C_M$ is self-adjoint operator, so we have

$$
\|(C_M + \lambda I)^{-1}(C_M - C_{M,\mathcal{D}})\| = \|(C_M + \lambda I)^{-1/2}(C_M - C_{M,\mathcal{D}})(C_M + \lambda I)^{-1/2}\|.
$$

According to Lemma 8 with $\upsilon_i = \phi_M(\mathbf{x}_i)$, we can obtain this result. □

**Proposition 7.** *If* $|\mathcal{D}| \geq 32\log(2/\delta)(1 + \mathcal{N}_\infty(\lambda))$, *then for any* $\delta > 0$, *with probability at least* $1 - \delta$, *we have*

$$
\mathcal{J}_{M,\mathcal{D}} := \|(C_{M,\mathcal{D}} + \lambda I)^{-1/2}(C_M + \lambda I)^{1/2}\| \leq \sqrt{2}.
$$

*Proof.* From Proposition 6, we know that if $|\mathcal{D}| \geq 32\log(2/\delta)(1 + \mathcal{N}_\infty(\lambda))$, then

$$
\|(C_M + \lambda I)^{-1/2}(C_{M,\mathcal{D}} - C_M)(C_M + \lambda)^{-1/2}\| \leq \frac{1}{2}.
$$

Combining the above inequality and Lemma 6, we can prove this result. □

**Proposition 8.** *If* $\delta \in (0, 1]$, *and* $|\mathcal{D}| \geq \Omega(\mathcal{N}_\infty(\lambda))$, *then with* $1 - \delta$, *we have*

$$
\|f_{M,\mathcal{D},\lambda} - f_{M,\lambda}\|_\rho = \mathcal{O}\left( \Upsilon_{M,\mathcal{D},\lambda}\log\frac{1}{\delta} + \|f_{M,\lambda} - f_\lambda\|_\rho + \|f_\lambda - f_\rho\|_\rho \right),
$$

*where* $\Upsilon_{M,\mathcal{D},\lambda} := \frac{1}{\sqrt{\lambda}|\mathcal{D}|} + \sqrt{\frac{\mathcal{N}(\lambda)}{|\mathcal{D}|}}$.

*Proof.* From Proposition 4, we have

$$
\|f_{M,\mathcal{D},\lambda} - f_{M,\lambda}\|_\rho \leq \mathcal{J}_{M,\mathcal{D}}^2(\mathcal{R}_{M,\mathcal{D}} + \mathcal{K}_{M,\mathcal{D}}) + (\mathcal{J}_{M,\mathcal{D}} + \mathcal{J}_{M,\mathcal{D}}^2)\|f_{M,\lambda} - f_\rho\|_\rho.
$$

Thus, from Lemmas 4, 5, and Propositions 5, 7, we know that if $|\mathcal{D}| \geq \Omega(\mathcal{N}_\infty(\lambda))$, we can prove this result. □

*Proof of Theorem 4.* According to Proposition 1, we have

$$
\begin{aligned}
\mathbb{E}\left[\|\bar{f}_{M,\mathcal{D},\lambda}^0 - f_\rho\|_\rho^2\right] &\leq 3\left[\|f_{M,\lambda} - f_\lambda\|_\rho^2\right] + 3\left[\|f_\lambda - f_\rho\|_\rho^2\right] \\
&+ 3\sum_{j=1}^m \frac{|\mathcal{D}_j|^2}{|\mathcal{D}|^2}\mathbb{E}\left[\|f_{M,\mathcal{D}_j,\lambda} - f_{M,\lambda}\|_\rho^2\right] + 3\sum_{j=1}^m \frac{|\mathcal{D}_j|}{|\mathcal{D}|}\mathbb{E}\left[\|f_{M,\mathcal{D}_j,\lambda}^\diamond - f_{M,\lambda}\|_\rho^2\right].
\end{aligned}
$$

Substituting Lemmas 2,3, Proposition 3, 7, 8 into the above inequality, one can see that if

$$
M \geq \Omega\left( \left( \frac{\mathcal{N}^{2r-1}(\lambda)}{\lambda^{2r-1}} \right)\mathcal{N}_\infty(\lambda)^{2-2r} \vee \mathcal{N}_\infty(\lambda) \right), \text{ and } |\mathcal{D}_j| \geq \Omega(\mathcal{N}_\infty(\lambda)),
$$

with confidence $1 - \delta$, we have

$$
\mathbb{E}\left[\|\bar{f}_{M,\mathcal{D},\lambda}^0 - f_\rho\|_\rho^2\right] = \mathcal{O}\left( \lambda^{2r} + \sum_{j=1}^m \frac{|\mathcal{D}_j|^2}{|\mathcal{D}|^2}\left( \Upsilon_{M,\mathcal{D}_j,\lambda}^2 \log^2\frac{1}{\delta} \right) + \sum_{j=1}^m \frac{|\mathcal{D}_j|}{|\mathcal{D}|}\lambda^{2r} \right), \tag{23}
$$

where $\Upsilon_{M,\mathcal{D}_j,\lambda} = \frac{1}{\sqrt{\lambda}|\mathcal{D}_j|} + \sqrt{\frac{\mathcal{N}(\lambda)}{|\mathcal{D}_j|}}$. If setting $|\mathcal{D}_1| = \ldots = |\mathcal{D}_m|$, $\lambda = \Omega(|\mathcal{D}|^{-\frac{1}{2r+\gamma}})$, we have

$$\Upsilon_{M,\mathcal{D}_j,\lambda} = \mathcal{O}\left(\sqrt{\frac{\mathcal{N}_M(\lambda)}{|\mathcal{D}_j|}} + \frac{1}{|\mathcal{D}_j|\sqrt{\lambda}}\right) = \mathcal{O}\left(\sqrt{m}|\mathcal{D}|^{-\frac{r}{2r+\gamma}} + m|\mathcal{D}|^{-\frac{4r+2\gamma-1}{4r+2\gamma}}\right). \tag{24}$$

Note that if $m \leq \Omega\left(|\mathcal{D}|^{\frac{2r+\gamma-1}{2r+\gamma}}\right)$ and $|\mathcal{D}_1| = \ldots = |\mathcal{D}_m|$, and $\lambda = \Omega(|\mathcal{D}|^{-\frac{1}{2r+\gamma}})$, we have

$$|\mathcal{D}_j| = \frac{|\mathcal{D}|}{m} \geq \Omega\left(|\mathcal{D}|^{\frac{1}{2r+\gamma}}\right) = \Omega(\mathcal{N}_\infty(\lambda)).$$

Thus, substituting 24 into 23, one can see that if $m \leq \Omega\left(|\mathcal{D}|^{\frac{2r+\gamma-1}{2r+\gamma}}\right)$ and

$$M \geq \Omega\left(\left(\left(\frac{\mathcal{N}^{2r-1}(\lambda)}{\lambda^{2r-1}}\right)\mathcal{N}_\infty(\lambda)^{2-2r} \vee \mathcal{N}_\infty(\lambda)\right) = \Omega\left(|\mathcal{D}|^{\frac{1+(2r-1)\gamma}{2r+\gamma}}\right),$$

with probability at least $1 - \delta$, we have

$$\mathbb{E}\left[\left\|\bar{f}_{M,\mathcal{D},\lambda}^0 - f_\rho\right\|_\rho^2\right] = \mathcal{O}\left(|\mathcal{D}|^{-\frac{2r}{2r+\gamma}}\log^2\frac{1}{\delta} + |\mathcal{D}|^{-1}\log^2\frac{1}{\delta} + |\mathcal{D}|^{-\frac{2r}{2r+\gamma}}\right)$$

$$= \mathcal{O}\left(|\mathcal{D}|^{-\frac{2r}{2r+\gamma}}\log^2\frac{1}{\delta}\right).$$

$\square$

# D APPENDIX: PROOF OF THEOREM 5

## D.1 APPENDIX: ERROR DECOMPOSITION FOR DKRR-RF IN PROBABILITY

**Proposition 9.** *The follows hold:*

$$\|\bar{\mathbf{w}}_{M,\mathcal{D},\lambda}^0 - \mathbf{w}_{M,\mathcal{D},\lambda}\| \leq \sum_{j=1}^m \frac{|\mathcal{D}_j|}{|\mathcal{D}|}\mathcal{J}_{M,\mathcal{D}}^2(\mathcal{Q}_{M,\mathcal{D}} + \mathcal{Q}_{M,\mathcal{D}_j})\|\mathbf{w}_{M,\mathcal{D}_j,\lambda} - \mathbf{w}_{M,\lambda}\| \text{ and}$$

$$\|\bar{f}_{M,\mathcal{D},\lambda}^0 - f_{M,\mathcal{D},\lambda}\| \leq \sum_{j=1}^m \frac{|\mathcal{D}_j|}{|\mathcal{D}|}\mathcal{J}_{M,\mathcal{D}}^2\left(\mathcal{Q}_{M,\mathcal{D}} + \mathcal{Q}_{M,\mathcal{D}_j}\right)$$

$$\left(\|f_{M,\mathcal{D}_j,\lambda} - f_{M,\lambda}\|_\rho + \sqrt{\lambda}\|\mathbf{w}_{M,\mathcal{D}_j,\lambda} - \mathbf{w}_{M,\lambda}\|\right),$$

*where* $\mathcal{J}_{M,\mathcal{D}} := \|(C_{M,\mathcal{D}} + \lambda I)^{-1/2}(C_M + \lambda I)^{1/2}\|$ *and* $\mathcal{Q}_{M,\mathcal{D}} := \|(C_M + \lambda I)^{-1/2}(C_M - C_{M,\mathcal{D}})(C_M + \lambda)^{-1/2}\|$.

*Proof.* Note that $\mathbf{w}_{M,\mathcal{D},\lambda} = (C_{M,\mathcal{D}} + \lambda I)^{-1} \mathbf{\Phi}_{M,\mathcal{D}} \bar{\mathbf{y}}_{\mathcal{D}}$, thus we have

$$
\begin{aligned}
&\bar{\mathbf{w}}^0_{M,\mathcal{D},\lambda} - \mathbf{w}_{M,\mathcal{D},\lambda} \\
&= \sum_{j=1}^{m} \frac{|\mathcal{D}_j|}{|\mathcal{D}|} (C_{M,\mathcal{D}_j} + \lambda I)^{-1} \mathbf{\Phi}_{M,\mathcal{D}_j} \bar{\mathbf{y}}_{\mathcal{D}_j} - (C_{M,\mathcal{D}} + \lambda I)^{-1} \mathbf{\Phi}_{M,\mathcal{D}} \bar{\mathbf{y}}_{\mathcal{D}} \\
&= \sum_{j=1}^{m} \frac{|\mathcal{D}_j|}{|\mathcal{D}|} \left( (C_{M,\mathcal{D}_j} + \lambda I)^{-1} - (C_{M,\mathcal{D}} + \lambda I)^{-1} \right) \mathbf{\Phi}_{M,\mathcal{D}_j} \bar{\mathbf{y}}_{\mathcal{D}_j} \\
&= \sum_{j=1}^{m} \frac{|\mathcal{D}_j|}{|\mathcal{D}|} (C_{M,\mathcal{D}} + \lambda I)^{-1} \left( C_{M,\mathcal{D}} - C_{M,\mathcal{D}_j} \right) (C_{M,\mathcal{D}_j} + \lambda I)^{-1} \mathbf{\Phi}_{M,\mathcal{D}_j} \bar{\mathbf{y}}_{\mathcal{D}_j} \\
&= \sum_{j=1}^{m} \frac{|\mathcal{D}_j|}{|\mathcal{D}|} (C_{M,\mathcal{D}} + \lambda I)^{-1} \left( C_{M,\mathcal{D}} - C_{M,\mathcal{D}_j} \right) \mathbf{w}_{M,\mathcal{D}_j,\lambda} \\
&= \sum_{j=1}^{m} \frac{|\mathcal{D}_j|}{|\mathcal{D}|} (C_{M,\mathcal{D}} + \lambda I)^{-1} \left( C_{M,\mathcal{D}} - C_M \right) \mathbf{w}_{M,\mathcal{D}_j,\lambda} \\
&\quad + \sum_{j=1}^{m} \frac{|\mathcal{D}_j|}{|\mathcal{D}|} (C_{M,\mathcal{D}} + \lambda I)^{-1} \left( C_M - C_{M,\mathcal{D}_j} \right) \mathbf{w}_{M,\mathcal{D}_j,\lambda} \\
&= \sum_{j=1}^{m} \frac{|\mathcal{D}_j|}{|\mathcal{D}|} (C_{M,\mathcal{D}} + \lambda I)^{-1} \left( C_{M,\mathcal{D}} - C_M \right) \left( \mathbf{w}_{M,\mathcal{D}_j,\lambda} - \mathbf{w}_{M,\lambda} \right) \\
&\quad + \sum_{j=1}^{m} \frac{|\mathcal{D}_j|}{|\mathcal{D}|} (C_{M,\mathcal{D}} + \lambda I)^{-1} \left( C_{M,\mathcal{D}} - C_M \right) \mathbf{w}_{M,\lambda} \\
&\quad + \sum_{j=1}^{m} \frac{|\mathcal{D}_j|}{|\mathcal{D}|} (C_{M,\mathcal{D}} + \lambda I)^{-1} \left( C_M - C_{M,\mathcal{D}_j} \right) \mathbf{w}_{M,\mathcal{D}_j,\lambda} \\
&= \sum_{j=1}^{m} \frac{|\mathcal{D}_j|}{|\mathcal{D}|} (C_{M,\mathcal{D}} + \lambda I)^{-1} \left( C_{M,\mathcal{D}} - C_M \right) \left( \mathbf{w}_{M,\mathcal{D}_j,\lambda} - \mathbf{w}_{M,\lambda} \right) \\
&\quad + \sum_{j=1}^{m} \frac{|\mathcal{D}_j|}{|\mathcal{D}|} (C_{M,\mathcal{D}} + \lambda I)^{-1} \left( C_M - C_{M,\mathcal{D}_j} \right) \left( \mathbf{w}_{M,\mathcal{D}_j,\lambda} - \mathbf{w}_{M,\lambda} \right).
\end{aligned}
\tag{25}
$$

Note that

$$
\begin{aligned}
&\sum_{j=1}^{m} \frac{|\mathcal{D}_j|}{|\mathcal{D}|} (C_{M,\mathcal{D}} + \lambda I)^{-1} \left( C_{M,\mathcal{D}} - C_M \right) \left( \mathbf{w}_{M,\mathcal{D}_j,\lambda} - \mathbf{w}_{M,\lambda} \right) \\
&= \sum_{j=1}^{m} \frac{|\mathcal{D}_j|}{|\mathcal{D}|} (C_{M,\mathcal{D}} + \lambda I)^{-1} (C_M + \lambda I)(C_M + \lambda I)^{-1} \left( C_{M,\mathcal{D}} - C_M \right) \left( \mathbf{w}_{M,\mathcal{D}_j,\lambda} - \mathbf{w}_{M,\lambda} \right) \\
\text{and } &\sum_{j=1}^{m} \frac{|\mathcal{D}_j|}{|\mathcal{D}|} (C_{M,\mathcal{D}} + \lambda I)^{-1} \left( C_M - C_{M,\mathcal{D}_j} \right) \left( \mathbf{w}_{M,\mathcal{D}_j,\lambda} - \mathbf{w}_{M,\lambda} \right) \\
&= \sum_{j=1}^{m} \frac{|\mathcal{D}_j|}{|\mathcal{D}|} (C_{M,\mathcal{D}} + \lambda I)^{-1} (C_M + \lambda I)(C_M + \lambda I)^{-1} \left( C_M - C_{M,\mathcal{D}_j} \right) \left( \mathbf{w}_{M,\mathcal{D}_j,\lambda} - \mathbf{w}_{M,\lambda} \right).
\end{aligned}
$$

Substituting the above equations into Eq. 25, we have

$$
\| \bar{\mathbf{w}}^0_{M,\mathcal{D},\lambda} - \mathbf{w}_{M,\mathcal{D},\lambda} \| \leq \sum_{j=1}^{m} \frac{|\mathcal{D}_j|}{|\mathcal{D}|} \mathcal{J}^2_{M,\mathcal{D}} (\mathcal{L}_{M,\mathcal{D}} + \mathcal{L}_{M,\mathcal{D}_j}) \| \mathbf{w}_{M,\mathcal{D}_j,\lambda} - \mathbf{w}_{M,\lambda} \|,
$$

where $\mathcal{L}_{M,\mathcal{D}} := \|(C_M + \lambda I)^{-1}(C_M - C_{M,\mathcal{D}})\|$. From Proposition 6, we know that $\mathcal{L}_{M,\mathcal{D}} = \mathcal{Q}_{M,\mathcal{D}}$, so we have

$$\|\bar{\mathbf{w}}^0_{M,\mathcal{D},\lambda} - \mathbf{w}_{M,\mathcal{D},\lambda}\| \leq \sum_{j=1}^{m} \frac{|\mathcal{D}_j|}{|\mathcal{D}|} \mathcal{J}^2_{M,\mathcal{D}} (\mathcal{Q}_{M,\mathcal{D}} + \mathcal{Q}_{M,\mathcal{D}_j}) \|\mathbf{w}_{M,\mathcal{D}_j,\lambda} - \mathbf{w}_{M,\lambda}\|,$$

which prove the first result of this proposition.

In the following, we will prove the second result of this proposition. Note that $S_M(\bar{\mathbf{w}}^0_{M,\mathcal{D},\lambda} - \mathbf{w}_{M,\mathcal{D},\lambda}) = \bar{f}^0_{M,\mathcal{D},\lambda} - f_{M,\mathcal{D},\lambda}$. According to 25, we have

$$\bar{f}^0_{M,\mathcal{D},\lambda} - f_{M,\mathcal{D},\lambda} = \sum_{j=1}^{m} \frac{|\mathcal{D}_j|}{|\mathcal{D}|} S_M(C_{M,\mathcal{D}} + \lambda I)^{-1} (C_{M,\mathcal{D}} - C_M) (\mathbf{w}_{M,\mathcal{D}_j,\lambda} - \mathbf{w}_{M,\lambda})$$

$$+ \sum_{j=1}^{m} \frac{|\mathcal{D}_j|}{|\mathcal{D}|} S_M(C_{M,\mathcal{D}} + \lambda I)^{-1} (C_M - C_{M,\mathcal{D}_j}) (\mathbf{w}_{M,\mathcal{D}_j,\lambda} - \mathbf{w}_{M,\lambda}) \quad (26)$$

$$:= \sum_{j=1}^{m} \frac{|\mathcal{D}_j|}{|\mathcal{D}|} (\hbar^1_j + \hbar^2_j).$$

Note that

$$\hbar^1_j = S_M(C_M + \lambda I)^{-1/2}(C_M + \lambda I)^{1/2}(C_{M,\mathcal{D}} + \lambda I)^{-1/2}(C_{M,\mathcal{D}} + \lambda I)^{-1/2}(C_M + \lambda I)^{1/2}$$

$$(C_M + \lambda I)^{-1/2} (C_{M,\mathcal{D}} - C_M) (C_M + \lambda I)^{-1/2} \quad (27)$$

$$(C_M + \lambda I)^{-1/2}(C_M + \lambda I)(\mathbf{w}_{M,\mathcal{D}_j,\lambda} - \mathbf{w}_{M,\lambda}),$$

thus we have

$$\|\hbar^1_j\|_\rho \leq \mathcal{J}^2_{M,\mathcal{D}} \mathcal{Q}_{M,\mathcal{D}} \|S_M(C_M + \lambda I)^{-1/2}\| \|(C_M + \lambda I)^{-1/2}(C_M + \lambda I)(\mathbf{w}_{M,\mathcal{D}_j,\lambda} - \mathbf{w}_{M,\lambda})\|$$

$$\leq \mathcal{J}^2_{M,\mathcal{D}} \mathcal{Q}_{M,\mathcal{D}} \|(C_M + \lambda I)^{-1/2}(C_M + \lambda I)(\mathbf{w}_{M,\mathcal{D}_j,\lambda} - \mathbf{w}_{M,\lambda})\|.$$

$$(28)$$

Since $\|S_M(C_M + \lambda I)^{-1/2}\| = \|(C_M + \lambda I)^{-1/2}C_M(C_M + \lambda I)^{-1/2}\|^{1/2} \leq 1$. So, we have

$$\|\hbar^1_j\|_\rho \leq \mathcal{J}^2_{M,\mathcal{D}} \mathcal{Q}_{M,\mathcal{D}} \|(C_M + \lambda I)^{-1/2}(C_M + \lambda I)(\mathbf{w}_{M,\mathcal{D}_j,\lambda} - \mathbf{w}_{M,\lambda})\|$$

$$= \mathcal{J}^2_{M,\mathcal{D}} \mathcal{Q}_{M,\mathcal{D}} \|(C_M + \lambda I)^{-1/2}(S^*_M S_M + \lambda I)(\mathbf{w}_{M,\mathcal{D}_j,\lambda} - \mathbf{w}_{M,\lambda})\|$$

$$\leq \mathcal{J}^2_{M,\mathcal{D}} \mathcal{Q}_{M,\mathcal{D}} \|(C_M + \lambda I)^{-1/2} S^*_M S_M (\mathbf{w}_{M,\mathcal{D}_j,\lambda} - \mathbf{w}_{M,\lambda})\|$$

$$+ \lambda \mathcal{J}^2_{M,\mathcal{D}} \mathcal{Q}_{M,\mathcal{D}} \|(C_M + \lambda I)^{-1/2}(\mathbf{w}_{M,\mathcal{D}_j,\lambda} - \mathbf{w}_{M,\lambda})\|$$

$$\leq \mathcal{J}^2_{M,\mathcal{D}} \mathcal{Q}_{M,\mathcal{D}} \|(C_M + \lambda I)^{-1/2} S^*_M\| \|f_{M,\mathcal{D}_j,\lambda} - f_{M,\mathcal{D},\lambda}\|_\rho$$

$$+ \sqrt{\lambda} \mathcal{J}^2_{M,\mathcal{D}} \mathcal{Q}_{M,\mathcal{D}} \|(\mathbf{w}_{M,\mathcal{D}_j,\lambda} - \mathbf{w}_{M,\lambda})\|$$

$$\leq \mathcal{J}^2_{M,\mathcal{D}} \mathcal{Q}_{M,\mathcal{D}} (\|f_{M,\mathcal{D}_j,\lambda} - f_{M,\mathcal{D},\lambda}\|_\rho + \sqrt{\lambda} \|\mathbf{w}_{M,\mathcal{D}_j,\lambda} - \mathbf{w}_{M,\lambda}\|),$$

the last inequality uses the fact that

$$\|(C_M + \lambda I)^{-1/2} S^*_M\| = \|(C_M + \lambda I)^{-1/2} S^*_M S_M (C_M + \lambda I)^{-1/2}\|^{1/2} \leq 1.$$

Similar as the above process, we can also obtain that

$$\|\hbar^2_j\|_\rho \leq \mathcal{J}^2_{M,\mathcal{D}} \mathcal{Q}_{M,\mathcal{D}_j} (\|f_{M,\mathcal{D}_j,\lambda} - f_{M,\mathcal{D},\lambda}\|_\rho - \sqrt{\lambda} \|\mathbf{w}_{M,\mathcal{D}_j,\lambda} - \mathbf{w}_{M,\lambda}\|).$$

Thus, using 26, we can prove this result. $\qquad\square$

### D.2 APPENDIX: PROOF OF THEOREM 5

***Proof of Theorem 5.*** Combining Proposition 9 and Proposition 4, we have

$$\|\bar{f}^0_{M,\mathcal{D},\lambda} - f_{M,\mathcal{D},\lambda}\|_\rho \leq \sum_{j=1}^{m} \frac{|\mathcal{D}_j|}{|\mathcal{D}|} \mathcal{J}^2_{M,\mathcal{D}} (\mathcal{Q}_{M,\mathcal{D}} + \mathcal{Q}_{M,\mathcal{D}_j})$$

$$\left( (\mathcal{J}_{M,\mathcal{D}_j} + \mathcal{J}^2_{M,\mathcal{D}_j})(\mathcal{R}_{M,\mathcal{D}_j} + \mathcal{K}_{M,\mathcal{D}_j}) + (2\mathcal{J}_{M,\mathcal{D}_j} + \mathcal{J}^2_{M,\mathcal{D}_j} + 1)\|f_{M,\lambda} - f_\rho\|_\rho \right). \quad (29)$$

From Propositions 7 and 8, one can see that if $|\mathcal{D}_j| \geq \Omega(\mathcal{N}_\infty(\lambda))$, and $\lambda \leq \|L_K\|$, then for any $\delta > 0$, with probability at least $1 - \delta$,

$$\|\bar{f}^0_{M,\mathcal{D},\lambda} - f_{M,\mathcal{D},\lambda}\|_\rho$$

$$=\mathcal{O}\left(\sum_{j=1}^m \frac{|\mathcal{D}_j|}{|\mathcal{D}|}\left(\mathcal{Q}_{M,\mathcal{D}} + \mathcal{Q}_{M,\mathcal{D}_j}\right)\left(\Upsilon_{M,\mathcal{D}_j,\lambda}\log\frac{1}{\delta} + \|f_{M,\lambda} - f_\lambda\|_\rho + \|f_\lambda - f_\rho\|_\rho\right)\right),$$

where $\Upsilon_{M,\mathcal{D}_i,\lambda} = \frac{1}{\sqrt{\lambda}|\mathcal{D}_i|} + \sqrt{\frac{\mathcal{N}(\lambda)}{|\mathcal{D}_i|}}$. Note that $\mathcal{Q}_{M,\mathcal{D}} \leq \mathcal{Q}_{M,\mathcal{D}_j}$, and by Lemmas 2 and 3, so we have

$$\|\bar{f}^0_{M,\mathcal{D},\lambda} - f_{M,\mathcal{D},\lambda}\|_\rho = \mathcal{O}\left(\sum_{j=1}^m \frac{|\mathcal{D}_j|}{|\mathcal{D}|}\mathcal{Q}_{M,\mathcal{D}_j}\Upsilon_{M,\mathcal{D}_j,\lambda}\log\frac{1}{\delta} + \lambda^r\mathcal{Q}_{M,\mathcal{D}_j}\right). \tag{30}$$

Note that

$$\|\bar{f}^0_{M,\mathcal{D},\lambda} - f_\rho\|_\rho = \|f^0_{M,\mathcal{D},\lambda} - f_{M,\mathcal{D},\lambda} + f_{M,\mathcal{D},\lambda} - f_{M,\lambda} + f_{M,\lambda} - f_\lambda + f_\lambda - f_\rho\|_\rho$$

$$\leq\|\bar{f}^0_{M,\mathcal{D},\lambda} - f_{M,\mathcal{D},\lambda}\|_\rho + \|f_{M,\mathcal{D},\lambda} - f_{M,\lambda}\|_\rho + \|f_{M,\lambda} - f_\lambda\|_\rho + \|f_\lambda - f\|_\rho. \tag{31}$$

Combining E.q 30, Lemmas 2, 3, Proposition 8 and E.q 31, one can see that if $M \geq \Omega\left(\left(\frac{\mathcal{N}^{2r-1}(\lambda)}{\lambda^{2r-1}}\right)\mathcal{N}_\infty(\lambda)^{2-2r} \vee \mathcal{N}_\infty(\lambda)\right)$, with probability $1 - \delta$, we have

$$\|\bar{f}^0_{M,\mathcal{D},\lambda} - f_\rho\|_\rho = \mathcal{O}\left(\sum_{j=1}^m \frac{|\mathcal{D}_j|}{|\mathcal{D}|}\mathcal{Q}_{M,\mathcal{D}_j}\Upsilon_{M,\mathcal{D}_j,\lambda}\log\frac{1}{\delta} + \Upsilon_{M,\mathcal{D},\lambda}\log\frac{1}{\delta} + \lambda^r\mathcal{Q}_{M,\mathcal{D}_j} + \lambda^r\right). \tag{32}$$

If $|\mathcal{D}_1| = \ldots = |\mathcal{D}_m|$, $\lambda = \Omega(|\mathcal{D}|^{-\frac{1}{2r+\gamma}})$, we have

$$\Upsilon_{M,\mathcal{D},\lambda} = \mathcal{O}\left(|\mathcal{D}|^{-\frac{r}{2r+\gamma}} + |\mathcal{D}|^{-\frac{4r+2\gamma-1}{4r+2\gamma}}\right) = \mathcal{O}\left(|\mathcal{D}|^{-\frac{r}{2r+\gamma}}\right),$$

$$\Upsilon_{M,\mathcal{D}_j,\lambda} = \mathcal{O}\left(\sqrt{m}|\mathcal{D}|^{-\frac{r}{2r+\gamma}} + m|\mathcal{D}|^{-\frac{4r+2\gamma-1}{4r+2\gamma}}\right), \tag{33}$$

$$\mathcal{Q}_{M,\mathcal{D}_j} = \mathcal{O}\left(m|\mathcal{D}|^{-\frac{2r+\gamma-1}{2r+\gamma}} + \sqrt{m}|\mathcal{D}|^{-\frac{2r+\gamma-1}{4r+2\gamma}}\right).$$

Thus, when $m \leq \Omega\left(|\mathcal{D}|^{\frac{2r+\gamma-1}{4r+2\gamma}}\right)$, we have

$$\Upsilon_{M,\mathcal{D}_j,\lambda}\mathcal{Q}_{M,\mathcal{D}_j} = \mathcal{O}\left(|\mathcal{D}|^{-\frac{r}{2r+\gamma}}\right),$$

$$\mathcal{Q}_{M,\mathcal{D}_j}\lambda^r = \mathcal{O}\left(|\mathcal{D}|^{-\frac{r}{2r+\gamma}}|\mathcal{D}|^{-\frac{2r-1+\gamma}{8r+4\gamma}}\right) = \mathcal{O}\left(|\mathcal{D}|^{-\frac{r}{2r+\gamma}}\right),$$

$$|\mathcal{D}_j| = \frac{|\mathcal{D}|}{m} \geq \Omega\left(|\mathcal{D}|^{\frac{2r+\gamma+1}{4r+2\gamma}}\right) \geq \Omega\left(|\mathcal{D}|^{\frac{1}{2r+\gamma}}\right) = \Omega(\mathcal{N}_\infty(\lambda)).$$

Thus, we have $\|\bar{f}^0_{M,\mathcal{D},\lambda} - f_\rho\|^2_\rho = \mathcal{O}\left(|\mathcal{D}|^{-\frac{2r}{2r+\gamma}}\log^2\frac{1}{\delta}\right)$, which prove this result. $\qquad\square$

# E   APPENDIX: PROOF OF THEOREM 6

## E.1   APPENDIX: ERROR DECOMPOSITION FOR DKRR-RF-CM IN PROBABILITY

**Proposition 10.**

$$\|\bar{f}^t_{M,\mathcal{D},\lambda} - f_{M,\mathcal{D},\lambda}\|_\rho$$

$$\leq\left(\sum_{i=1}^m \frac{|\mathcal{D}_j|}{|\mathcal{D}|}(2\mathcal{J}^2_{M,\mathcal{D}_j}\mathcal{Q}_{M,\mathcal{D}} + 2\mathcal{J}^2_{M,\mathcal{D}_j}\mathcal{Q}_{M,\mathcal{D}_j})\right)^t\left(\|\bar{f}^0_{M,\mathcal{D},\lambda} - f_{M,\mathcal{D},\lambda}\|_\rho + \sqrt{\lambda}\|\bar{\mathbf{w}}^0_{M,\mathcal{D},\lambda} - \mathbf{w}_{M,\mathcal{D},\lambda}\|\right),$$

where $\mathcal{J}_{M,\mathcal{D}} := \|(C_{M,\mathcal{D}} + \lambda I)^{-1/2}(C_M + \lambda I)^{1/2}\|$ and $\mathcal{Q}_{M,\mathcal{D}} := \|(C_M + \lambda I)^{-1/2}(C_M - C_{M,\mathcal{D}})(C_M + \lambda)^{-1/2}\|$.

*Proof.* Note that

$$\mathbf{w}_{M,\mathcal{D},\lambda} = \bar{\mathbf{w}}_{M,\mathcal{D},\lambda}^{t-1} - (C_{M,\mathcal{D}} + \lambda I)^{-1} \left[ (C_{M,\mathcal{D}} + \lambda I)\bar{\mathbf{w}}_{M,\mathcal{D},\lambda}^{t-1} - \boldsymbol{\Phi}_{M,\mathcal{D}}\bar{\mathbf{y}}_{\mathcal{D}} \right],$$

$$\bar{\mathbf{w}}_{M,\mathcal{D},\lambda}^{t} = \bar{\mathbf{w}}_{M,\mathcal{D},\lambda}^{t-1} - \sum_{j=1}^{m} \frac{|\mathcal{D}_j|}{|\mathcal{D}|}(C_{M,\mathcal{D}_j} + \lambda I)^{-1} \left[ (C_{M,\mathcal{D}} + \lambda I)\bar{\mathbf{w}}_{M,\mathcal{D},\lambda}^{t-1} - \boldsymbol{\Phi}_{M,\mathcal{D}}\bar{\mathbf{y}}_{\mathcal{D}} \right].$$

Thus, we have

$$\mathbf{w}_{M,\mathcal{D},\lambda} - \bar{\mathbf{w}}_{M,\mathcal{D},\lambda}^{t}$$

$$= \bar{\mathbf{w}}_{M,\mathcal{D},\lambda}^{t-1} - (C_{M,\mathcal{D}} + \lambda I)^{-1} \left[ (C_{M,\mathcal{D}} + \lambda I)\bar{\mathbf{w}}_{M,\mathcal{D},\lambda}^{t-1} - \boldsymbol{\Phi}_{M,\mathcal{D}}\bar{\mathbf{y}}_{\mathcal{D}} \right]$$

$$\quad - \bar{\mathbf{w}}_{M,\mathcal{D},\lambda}^{t-1} + \sum_{j=1}^{m} \frac{|\mathcal{D}_j|}{|\mathcal{D}|}(C_{M,\mathcal{D}_j} + \lambda I)^{-1} \left[ (C_{M,\mathcal{D}} + \lambda I)\bar{\mathbf{w}}_{M,\mathcal{D},\lambda}^{t-1} - \boldsymbol{\Phi}_{M,\mathcal{D}}\bar{\mathbf{y}}_{\mathcal{D}} \right]$$

$$= \sum_{j=1}^{m} \frac{|\mathcal{D}_j|}{|\mathcal{D}|} \left[ (C_{M,\mathcal{D}_j} + \lambda I)^{-1} - (C_{M,\mathcal{D}} + \lambda I)^{-1} \right] \left[ (C_{M,\mathcal{D}} + \lambda I)\bar{\mathbf{w}}_{M,\mathcal{D},\lambda}^{t-1} - \boldsymbol{\Phi}_{M,\mathcal{D}}\bar{\mathbf{y}}_{\mathcal{D}} \right]$$

$$= \sum_{j=1}^{m} \frac{|\mathcal{D}_j|}{|\mathcal{D}|}(C_{M,\mathcal{D}_j} + \lambda I)^{-1} \left[ C_{M,\mathcal{D}} - C_{M,\mathcal{D}_j} \right] (C_{M,\mathcal{D}} + \lambda I)^{-1} \left[ (C_{M,\mathcal{D}} + \lambda I)\bar{\mathbf{w}}_{M,\mathcal{D},\lambda}^{t-1} - \boldsymbol{\Phi}_{M,\mathcal{D}}\bar{\mathbf{y}}_{\mathcal{D}} \right]$$

$$= \sum_{j=1}^{m} \frac{|\mathcal{D}_j|}{|\mathcal{D}|}(C_{M,\mathcal{D}_j} + \lambda I)^{-1} \left[ C_{M,\mathcal{D}} - C_{M,\mathcal{D}_j} \right] [\bar{\mathbf{w}}_{M,\mathcal{D},\lambda}^{t-1} - \mathbf{w}_{M,\mathcal{D},\lambda}]$$

$$= \sum_{j=1}^{m} \frac{|\mathcal{D}_j|}{|\mathcal{D}|}(C_{M,\mathcal{D}_j} + \lambda I)^{-1} \left[ C_{M,\mathcal{D}} - C_M \right] [\bar{\mathbf{w}}_{M,\mathcal{D},\lambda}^{t-1} - \mathbf{w}_{M,\mathcal{D},\lambda}]$$

$$\quad + \sum_{j=1}^{m} \frac{|\mathcal{D}_j|}{|\mathcal{D}|}(C_{M,\mathcal{D}_j} + \lambda I)^{-1} \left[ C_M - C_{M,\mathcal{D}_j} \right] [\bar{\mathbf{w}}_{M,\mathcal{D},\lambda}^{t-1} - \mathbf{w}_{M,\mathcal{D},\lambda}]$$

$$:= \sum_{j=1}^{m} \frac{|\mathcal{D}_j|}{|\mathcal{D}|}\aleph_1^j + \aleph_2^j.$$

(34)

Note that

$$S_M\aleph_1^j = S_M(C_M + \lambda I)^{-1/2}(C_M + \lambda I)^{1/2}(C_{M,\mathcal{D}_j} + \lambda I)^{-1/2}$$
$$(C_{M,\mathcal{D}_j} + \lambda I)^{-1/2}(C_M + \lambda I)^{1/2}(C_M + \lambda I)^{-1/2} \left[ C_{M,\mathcal{D}} - C_M \right] (C_M + \lambda I)^{-1/2} \quad (35)$$
$$(C_M + \lambda I)^{-1/2}(C_M + \lambda I) \left( \bar{\mathbf{w}}_{M,\mathcal{D},\lambda}^{t-1} - \mathbf{w}_{M,\mathcal{D},\lambda} \right).$$

Note that $\|S_M(C_M + \lambda I)^{-1/2}\| = \|(C_M + \lambda I)^{-1/2}C_M(C_M + \lambda I)^{-1/2}\|^{1/2} \leq 1$, so, we have

$$\|S_M\aleph_1^j\|_\rho \leq \mathcal{J}_{M,\mathcal{D}_j}^2 \mathcal{Q}_{M,\mathcal{D}} \left\| (C_M + \lambda I)^{-1/2}(C_M + \lambda I) \left( \bar{\mathbf{w}}_{M,\mathcal{D},\lambda}^{t-1} - \mathbf{w}_{M,\mathcal{D},\lambda} \right) \right\|. \quad (36)$$

Note that $C_M = S_M^* S_M$, so

$$C_M \left( \bar{\mathbf{w}}_{M,\mathcal{D},\lambda}^{t-1} - \mathbf{w}_{M,\mathcal{D},\lambda} \right) = S_M^* S_M \left( \bar{\mathbf{w}}_{M,\mathcal{D},\lambda}^{t-1} - \mathbf{w}_{M,\mathcal{D},\lambda} \right) = S_M^* \left( \bar{f}_{M,\mathcal{D},\lambda}^{t-1} - f_{M,\mathcal{D},\lambda} \right).$$

Substituting the above inequality into Eq. 36, we have

$$\|S_M\aleph_1^j\|_\rho \leq \mathcal{J}_{M,\mathcal{D}_j}^2 \mathcal{Q}_{M,\mathcal{D}} \left\| (C_M + \lambda I)^{-1/2}S_M^* \left( \bar{f}_{M,\mathcal{D},\lambda}^{t-1} - f_{M,\mathcal{D},\lambda} \right) \right\|$$

$$\quad + \lambda \mathcal{J}_{M,\mathcal{D}_j}^2 \mathcal{Q}_{M,\mathcal{D}} \left\| (C_M + \lambda I)^{-1/2} \left( \bar{\mathbf{w}}_{M,\mathcal{D},\lambda}^{t-1} - \mathbf{w}_{M,\mathcal{D},\lambda} \right) \right\|$$

$$\quad \leq \mathcal{J}_{M,\mathcal{D}_j}^2 \mathcal{Q}_{M,\mathcal{D}} \left( \|\bar{f}_{M,\mathcal{D},\lambda}^{t-1} - f_{M,\mathcal{D},\lambda}\|_\rho + \sqrt{\lambda}\|\bar{\mathbf{w}}_{M,\mathcal{D},\lambda}^{t-1} - \mathbf{w}_{M,\mathcal{D},\lambda}\| \right),$$

the last inequality use the fact that $\|(C_M + \lambda I)^{-1/2} S_M^*\| = \|(C_M + \lambda I)^{-1/2} C_M (C_M + \lambda I)^{-1/2}\|^{1/2} \leq 1$. Using the same process, we can obtain that

$$\|S_M \aleph_2^j\|_\rho \leq \mathcal{J}_{M,\mathcal{D}_j}^2 \mathcal{Q}_{M,\mathcal{D}_j} \left( \|\bar{f}_{M,\mathcal{D},\lambda}^{t-1} - f_{M,\mathcal{D},\lambda}\|_\rho + \sqrt{\lambda} \|\bar{\mathbf{w}}_{M,\mathcal{D},\lambda}^{t-1} - \mathbf{w}_{M,\mathcal{D},\lambda}\| \right).$$

Thus, we have

$$\|f_{M,\mathcal{D},\lambda} - \bar{f}_{M,\mathcal{D},\lambda}^t\|_\rho = \|S_M(\mathbf{w}_{M,\mathcal{D},\lambda} - \bar{\mathbf{w}}_{M,\mathcal{D},\lambda}^t)\|_\rho \leq \sum_{j=1}^m \frac{|\mathcal{D}_j|}{|\mathcal{D}|} \|S_M \aleph_1^j\|_\rho + \|S_M \aleph_2^j\|_\rho$$

$$\leq \sum_{j=1}^m \frac{|\mathcal{D}_j|}{|\mathcal{D}|} (\mathcal{J}_{M,\mathcal{D}_j}^2 \mathcal{Q}_{M,\mathcal{D}} + \mathcal{J}_{M,\mathcal{D}_j}^2 \mathcal{Q}_{M,\mathcal{D}_j}) \quad (37)$$

$$\left( \|\bar{f}_{M,\mathcal{D},\lambda}^{t-1} - f_{M,\mathcal{D},\lambda}\|_\rho + \sqrt{\lambda} \|\bar{\mathbf{w}}_{M,\mathcal{D},\lambda}^{t-1} - \mathbf{w}_{M,\mathcal{D},\lambda}\| \right).$$

According to 34, we know that

$$\mathbf{w}_{M,\mathcal{D},\lambda} - \bar{\mathbf{w}}_{M,\mathcal{D},\lambda}^t = \sum_{j=1}^m \frac{|\mathcal{D}_j|}{|\mathcal{D}|} (\aleph_1^j + \aleph_2^j)$$

$$= \sum_{j=1}^m \frac{|\mathcal{D}_j|}{|\mathcal{D}|} (C_{M,\mathcal{D}_j} + \lambda I)^{-1} [C_{M,\mathcal{D}} - C_M] [\bar{\mathbf{w}}_{M,\mathcal{D},\lambda}^{t-1} - \mathbf{w}_{M,\mathcal{D},\lambda}]$$

$$+ \sum_{j=1}^m \frac{|\mathcal{D}_j|}{|\mathcal{D}|} (C_{M,\mathcal{D}_j} + \lambda I)^{-1} [C_M - C_{M,\mathcal{D}_j}] [\bar{\mathbf{w}}_{M,\mathcal{D},\lambda}^{t-1} - \mathbf{w}_{M,\mathcal{D},\lambda}]$$

$$= \sum_{j=1}^m \frac{|\mathcal{D}_j|}{|\mathcal{D}|} (C_{M,\mathcal{D}_j} + \lambda I)^{-1} (C_M + \lambda I)(C_M + \lambda I)^{-1} [C_{M,\mathcal{D}} - C_M] [\bar{\mathbf{w}}_{M,\mathcal{D},\lambda}^{t-1} - \mathbf{w}_{M,\mathcal{D},\lambda}]$$

$$+ \sum_{j=1}^m \frac{|\mathcal{D}_j|}{|\mathcal{D}|} (C_{M,\mathcal{D}_j} + \lambda I)^{-1} (C_M + \lambda I)(C_M + \lambda I)^{-1} [C_M - C_{M,\mathcal{D}_j}] [\bar{\mathbf{w}}_{M,\mathcal{D},\lambda}^{t-1} - \mathbf{w}_{M,\mathcal{D},\lambda}].$$

Thus, one can obtain that

$$\|\mathbf{w}_{M,\mathcal{D},\lambda} - \bar{\mathbf{w}}_{M,\mathcal{D},\lambda}^t\| \leq \sum_{j=1}^m \mathcal{J}_{M,\mathcal{D}_j}^2 (\mathcal{L}_{M,\mathcal{D}} + \mathcal{L}_{M,\mathcal{D}_j}) \|\bar{\mathbf{w}}_{M,\mathcal{D},\lambda}^{t-1} - \mathbf{w}_{M,\mathcal{D},\lambda}\|,$$

where $\mathcal{L}_{M,\mathcal{D}} := \|(C_M + \lambda I)^{-1}(C_M - C_{M,\mathcal{D}})\|$. From Proposition 6, we know that $\mathcal{L}_{M,\mathcal{D}} = \mathcal{Q}_{M,\mathcal{D}}$. Thus, we have

$$\|\mathbf{w}_{M,\mathcal{D},\lambda} - \bar{\mathbf{w}}_{M,\mathcal{D},\lambda}^t\| \leq \sum_{j=1}^m \mathcal{J}_{M,\mathcal{D}_j}^2 (\mathcal{Q}_{M,\mathcal{D}} + \mathcal{Q}_{M,\mathcal{D}_j}) \|\bar{\mathbf{w}}_{M,\mathcal{D},\lambda}^{t-1} - \mathbf{w}_{M,\mathcal{D},\lambda}\|. \quad (38)$$

Combining 37 and 38, we have

$$\|f_{M,\mathcal{D},\lambda} - \bar{f}_{M,\mathcal{D},\lambda}^t\|_\rho + \sqrt{\lambda} \|\mathbf{w}_{M,\mathcal{D},\lambda} - \bar{\mathbf{w}}_{M,\mathcal{D},\lambda}^t\|$$

$$\leq \sum_{j=1}^m \frac{|\mathcal{D}_j|}{|\mathcal{D}|} (\mathcal{J}_{M,\mathcal{D}_i}^2 \mathcal{Q}_{M,\mathcal{D}} + \mathcal{J}_{M,\mathcal{D}_j}^2 \mathcal{Q}_{M,\mathcal{D}_j}) \left( \|\bar{f}_{M,\mathcal{D},\lambda}^{t-1} - f_{M,\mathcal{D},\lambda}\|_\rho + \sqrt{\lambda} \|\bar{\mathbf{w}}_{M,\mathcal{D},\lambda}^{t-1} - \mathbf{w}_{M,\mathcal{D},\lambda}\| \right)$$

$$+ \sum_{j=1}^m \frac{|\mathcal{D}_j|}{|\mathcal{D}|} \mathcal{J}_{M,\mathcal{D}_j}^2 (\mathcal{Q}_{M,\mathcal{D}} + \mathcal{Q}_{M,\mathcal{D}_j}) \sqrt{\lambda} \|\bar{\mathbf{w}}_{M,\mathcal{D},\lambda}^{t-1} - \mathbf{w}_{M,\mathcal{D},\lambda}\|$$

$$\leq \sum_{j=1}^m \frac{|\mathcal{D}_j|}{|\mathcal{D}|} (2\mathcal{J}_{M,\mathcal{D}_j}^2 \mathcal{Q}_{M,\mathcal{D}_j} + 2\mathcal{J}_{M,\mathcal{D}_j}^2 \mathcal{Q}_{M,\mathcal{D}_j}) \left( \|\bar{f}_{M,\mathcal{D},\lambda}^{t-1} - f_{M,\mathcal{D},\lambda}\|_\rho + \sqrt{\lambda} \|\bar{\mathbf{w}}_{M,\mathcal{D},\lambda}^{t-1} - \mathbf{w}_{M,\mathcal{D},\lambda}\| \right)$$

$$\leq \left( 2 \sum_{j=1}^m \frac{|\mathcal{D}_j|}{|\mathcal{D}|} \mathcal{J}_{M,\mathcal{D}_j}^2 \mathcal{Q}_{M,\mathcal{D}} + \mathcal{J}_{M,\mathcal{D}_j}^2 \mathcal{Q}_{M,\mathcal{D}_j} \right)^t \left( \|\bar{f}_{M,\mathcal{D},\lambda}^0 - f_{M,\mathcal{D},\lambda}\|_\rho + \sqrt{\lambda} \|\bar{\mathbf{w}}_{M,\mathcal{D},\lambda}^0 - \mathbf{w}_{M,\mathcal{D},\lambda}\| \right),$$

which prove the result. $\qquad\square$

### E.2 APPENDIX: PROOF OF THEOREM 6

***Proof of Theorem 6.*** Substituting Propositions 4, 5, 7, 8 and Lemma 4 into Proposition 9, we have

$$\|\bar{f}_{M,\mathcal{D},\lambda}^0 - f_{M,\mathcal{D},\lambda}\|_\rho + \sqrt{\lambda}\|\bar{\mathbf{w}}_{\mathcal{D},\lambda}^0 - \mathbf{w}_{M,\mathcal{D},\lambda}\|_2$$

$$=\mathcal{O}\left(\sum_{j=1}^m \frac{|\mathcal{D}_j|}{|\mathcal{D}|}\left(\mathcal{Q}_{M,\mathcal{D}} + \mathcal{Q}_{M,\mathcal{D}_j}\right)\left(\mathcal{R}_{M,\mathcal{D}} + \mathcal{K}_{M,\mathcal{D}} + \|f_{M,\lambda} - f_\rho\|_\rho\right)\right)$$

$$=\mathcal{O}\left(\sum_{j=1}^m \frac{|\mathcal{D}_j|}{|\mathcal{D}|}\left(\mathcal{Q}_{M,\mathcal{D}_j} + \mathcal{Q}_{M,\mathcal{D}_j}\right)\left(\Upsilon_{M,\mathcal{D}_j,\lambda} + \|f_{M,\lambda} - f_\lambda\| + \|f_\lambda - f_\rho\|_\rho\right)\right).$$

Combining the above inequality and Proposition 10, and note that $\mathcal{Q}_{M,\mathcal{D}} \leq \mathcal{Q}_{M,\mathcal{D}_j}$, we can obtain that

$$\|\bar{f}_{M,\mathcal{D},\lambda}^t - f_{M,\mathcal{D},\lambda}\|_\rho$$

$$\leq\mathcal{O}\left(\left(\sum_{j=1}^m \frac{|\mathcal{D}_j|}{|\mathcal{D}|}\mathcal{Q}_{M,\mathcal{D}_j}\right)^t \left(\sum_{j=1}^m \frac{|\mathcal{D}_j|}{|\mathcal{D}|}\mathcal{Q}_{M,\mathcal{D}_j}\left(\Upsilon_{M,\mathcal{D}_j,\lambda} + \|f_{M,\lambda} - f_\lambda\| + \|f_\lambda - f_\rho\|_\rho\right)\right)\right).$$

Note that

$$\|\bar{f}_{M,\mathcal{D},\lambda}^t - f_\rho\|_\rho = \|f_{M,\mathcal{D},\lambda}^0 - f_{M,\mathcal{D},\lambda} + f_{M,\mathcal{D},\lambda} - f_{M,\lambda} + f_{M,\lambda} - f_\lambda + f_\lambda - f_\rho\|_\rho$$

$$\leq\|\bar{f}_{M,\mathcal{D},\lambda}^t - f_{M,\mathcal{D},\lambda}\|_\rho + \|f_{M,\mathcal{D},\lambda} - f_{M,\lambda}\|_\rho + \|f_{M,\lambda} - f_\lambda\|_\rho + \|f_\lambda - f\|_\rho. \qquad (39)$$

Thus, by Lemmas 2, 3, one can see that if

$$M \geq \Omega\left(\left(\frac{\mathcal{N}^{2r-1}(\lambda)}{\lambda^{2r-1}}\right)\mathcal{N}_\infty(\lambda)^{2-2r} \vee \mathcal{N}_\infty(\lambda)\right),$$

we have

$$\|\bar{f}_{M,\mathcal{D},\lambda}^t - f_\rho\|_\rho$$

$$=\mathcal{O}\left(\left(\sum_{j=1}^m \frac{|\mathcal{D}_j|}{|\mathcal{D}|}\mathcal{Q}_{M,\mathcal{D}_j}\right)^t \left(\sum_{j=1}^m \frac{|\mathcal{D}_j|}{|\mathcal{D}|}\mathcal{Q}_{M,\mathcal{D}_j}\left(\Upsilon_{M,\mathcal{D}_j,\lambda} + \lambda^r\right)\right) + \Upsilon_{M,\mathcal{D},\lambda}\log\frac{1}{\delta} + \lambda^r\right).$$

From 33, by setting $|\mathcal{D}_1| = \ldots = |\mathcal{D}_m|$, $\lambda = \Omega(|\mathcal{D}|^{-\frac{1}{2r+\gamma}})$, we know that

$$\Upsilon_{M,\mathcal{D}_j,\lambda} = \mathcal{O}\left(\sqrt{m}|\mathcal{D}|^{-\frac{r}{2r+\gamma}} + m|\mathcal{D}|^{-\frac{4r+2\gamma-1}{4r+2\gamma}}\right),$$

$$\mathcal{Q}_{M,\mathcal{D}_j} = \mathcal{O}\left(m|\mathcal{D}|^{-\frac{2r+\gamma-1}{2r+\gamma}} + \sqrt{m}|\mathcal{D}|^{-\frac{2r+\gamma-1}{4r+2\gamma}}\right),$$

$$\Upsilon_{M,\mathcal{D},\lambda} = \mathcal{O}\left(|\mathcal{D}|^{-\frac{r}{2r+\gamma}} + |\mathcal{D}|^{-\frac{4r+2\gamma-1}{4r+2\gamma}}\right).$$

Thus, when $m \leq \Omega\left(|\mathcal{D}|^{\frac{(2r+\gamma-1)(t+1)}{(2r+\gamma)(t+2)}}\right)$, we have $\|\bar{f}_{M,\mathcal{D},\lambda}^t - f_\rho\|_\rho = \mathcal{O}\left(|\mathcal{D}|^{-\frac{r}{2r+\gamma}}\right)$, which proves the result. $\qquad\square$

## F APPENDIX: PROOF OF THEOREMS 1, 2, 3

*Proof.* From (Smale & Zhou, 2007; Caponnetto & Vito, 2007), if $r = 1/2$, then $f_\rho \in \mathcal{H}_K$. Thus, in this case, $f_{\mathcal{H}_K}$ exists and $\mathcal{E}(f_{\mathcal{H}_K}) = \mathcal{E}(f_\rho)$. Note that Assumption 1 is always satisfied for $\gamma = 1$ and $c = \tau^2$. So, using Theorems 4, 5 and 6 with $r = 1/2$, $\gamma = 1$ and $c = \tau^2$, Theorem 1, 2, and 3 can be proved. $\qquad\square$

