# OpenReview forum: "Effective Distributed Learning with Random Features: Improved Bounds and Algorithms"
_ICLR.cc/2021/Conference — ICLR 2021 Poster_

### Official Review · AnonReviewer4 · 2020-10-27
**Non-trivial improvements combining distributed iterative learning and random features for KRR**

**Rating:** 6
**Confidence:** 3

**Review:**

The paper analyses generalization properties of distributed kernel ridge regression (DKRR) with random features and communications. It studies optimal learning rates of the generalization bounds both in expectation and in probability. In the case of DKRR with random features, the optimal learning rate in expectation is shown to achieve by relaxing the requirement on the number of partitions from $O(1)$ (Li et al., 2019a) to $O(|D|^{0.5})$ (Theorem 1). Within the same setup of random features, the number of partitions is relaxed to $O(|D|^{0.25})$ guaranteeing optimal generalization performance in probability (Theorem 2). The latter bound $O(|D|^{0.25})$ on partition count is much smaller then $O(|D|^{0.5})$. However, as proved in Theorem 3, allowing multiple communication rounds in DKRR-RF, up to $O(|D|^{0.5})$ partitions can be handled depending on the number of communication rounds. In other words, it can exploit more partitions at the cost of more communication rounds.

The idea of DKRR with random features and communications is a combination of DKRR with random features studied in (Li et al., 2019a) and DKRR with communications studied in (Lin et al., 2020). It seems that the communication strategy and Algorithm 1 presented in section 3 are adaptations from (Lin et al., 2020) handling random feature, which should be properly mentioned and credited.

During the comparison with (Lin et al., 2020), it is mentioned that the letter work required communicating local data $D_j$ among partition nodes. However, I failed to spot it in (Lin et al., 2020), and instead found similar steps of Algorithm 1 in section 2.3 of (Lin et al., 2020), where only gradient information is communicated. Please elaborate on this.

Not being an expert in this narrow field, I think the improvements are essential and would be helpful for the associated community. The paper is well written with fair amount of discussion comparing with the results and proof techniques of recent works.

---

> ### Author Response · Authors · 2020-11-13
> **Response to Reviewer4**
>
> Thanks for your recognition of this paper.
>
> In this paper, we extend the communication strategy proposed in (Lin et al., 2020) to random features to avoid communicating local data $\mathcal{D}_j$ among partition nodes. We have mentioned and credited this in the rebuttal revision. Thanks for your suggestion.
>
> In Section 2.3 of (Lin et al., 2020), it seems that it only need to communicate the gradient information, but it should be noted that the gradient information is based on an operator representation (see Eq.(7) in Lin et al., 2020), which is usually infeasible in practice. The authors present a realization for the proposed strategy by communicating the data among each local machine, see in Appendix B (page 34 in Lin et al., 2020, step 1). We have elaborated on this in the rebuttal revision.

---

### Official Review · AnonReviewer2 · 2020-10-28
**This paper studies the statistical properties of distributed kernel ridge regression together with random features (DKRR-RF), and obtain optimal generalization bounds under the basic setting in the attainable cases.  Numerical results are given for the studied new algorithms. However, the presentations as well as the citations need some major revision before the publication.**

**Rating:** 6
**Confidence:** 3

**Review:**

This paper studies the statistical properties of distributed kernel ridge regression together with random features (DKRR-RF), and obtain optimal generalization bounds under the basic setting in the attainable cases.  Numerical results are given for the studied new algorithms. The algorithms and the derived results are new and interesting to me. However, the presentations as well as the citations need some major revision before the publication.

Some few comments:
- It looks to me that the idea of distributed learning with communication has already appeared in [ arXiv:1906.04870] if not earlier.
-Page 1. The authors mention that distributed learning has been combined with multi-pass SGD, but they did not cite the related paper [Optimal Distributed Learning with Multi-pass Stochastic Gradient Methods, ICML 2018]
-Page 2. Optimal learning rates were also established for distributed
spectral algorithms in [JMLR 2018, 19(1): 1069-1097] ( or [arXiv:1610.07487]).
-Page 2 and the other locations.  Optimal learning rates with a less strict condition on the number of local machines were first established in [JMLR 2020, 21(147): 1-63] (or [arXiv:1801.07226]) if not earlier.
-Bottom of Page 6, to my knowledge,  the first one using the concentration inequality for self-adjoint operators to relax the restriction on the number of local machines is [JMLR 2020, 21(147): 1-63] (or [arXiv:1801.07226], or [Optimal Distributed Learning with Multi-pass Stochastic Gradient Methods, ICML 2018]) (if not earlier).
Also, the first part of Proposition 6 was first proved in [Random design analysis of ridge regression. 2012 COLT] for the matrix case, and later was extended to the operator case in [ On the sample complexity of subspace learning. NeurIPS 2013]
-Finite-sample theoretical analysis about the approximation quality of RFFs has been established in [On the error of random Fourier features. UAI 2015] and [Optimal Rates for Random Fourier Features, NeurIPS 2015.]
-Numerical results on different data-sets could be given to further exemplify the performance of the algorithm.
-How do you choose the regularization parameter $\lambda$ in the distributed learning? Will this enlarge the computational complexity?

---

> ### Author Response · Authors · 2020-11-13
> **Response to Reviewer2**
>
> Thanks for your recognition of this paper.
>
> We will make a major revision to improve our presentations as well as the citations. The research on the optimal learning of distributed learning is a fast-growing field. In this paper, we mainly focus on the distributed learning with random features, thus the single distributed learning may be ignored. In the rebuttal revision, we have added the reference [Optimal Distributed Learning with Multi-pass Stochastic Gradient Methods, ICML 2018] after the multi-pass SGD in page 1, added the reference about the distributed spectral algorithms [JMLR 2018, 19(1): 1069-1097] in page 2, and added other references [arXiv:1906.04870, JMLR 2020, 21(147): 1-63; Random design analysis of ridge regression, 2012 COLT; On the sample complexity of subspace learning, NeurIPS 2013; On the error of random Fourier features, UAI 2015;  Optimal Rates for Random Fourier Features, NeurIPS 2015] in the appropriate locations.  Thanks for your suggestions.
>
> Experiments on simulated datasets and real dataset (MINIST) have been provided to validate our theoretical findings. We will add more numerical results on different datasets to further exemplify the performance of the algorithm in the final version. In fact, we are doing experiments now, and hoping we can have the numerical results during the discussion phases. Thanks for your suggestion.
>
> According to Theorem 1,2,3, to guarantee the optimal rates, the regularization parameter $\lambda$ should be around $|\mathcal{D}|^{-1/2}$, thus in this paper, we consider using the 5-fold cross validation to fine-tune the $\lambda$, the tuned set is $\{2^{-5}|\mathcal{D}|^{-1/2},2^{-3}|\mathcal{D}|^{-1/2},\ldots,2^5|\mathcal{D}|^{-1/2}\}$. Please refer to the detail in the first part of page 8. Because of using the 5-fold cross validation in selecting the optimal $\lambda$, the computational complexity should be enlarged. However, it should be noted that even for the plain methods, tuning the optimal $\lambda$ is also required, which will enlarge the computational complexity as well. We have added above discussion in the rebuttal revision.

---

> ### Author Response · Authors · 2020-11-21
> **More numerical results on different data-sets are given.**
>
> Thanks for your recognition of this paper.
>
> A new revision of our submission are updated.
> More experiments results on different data-sets  are provided to further exemplify the performance of the proposed algorithm (see APPENDIX G). One can find that  our communication-based DKRR-RF are better than the original DKRR-RF on all data-sets,  which demonstrates the  effectiveness of our methods.

---

### Official Review · AnonReviewer1 · 2020-10-28
**ICLR 2021 Conference Paper1362 AnonReviewer1**

**Rating:** 4
**Confidence:** 4

**Review:**

The paper investigates an algorithm for distributed learning with random Fourier features. The main idea is to sample M random Fourier features and split the data into m chunks. Each chunk is processed on a separate machine that outputs a linear hypothesis using the sampled M random features. The hypotheses coming from different machines are then aggregated on the master machine via importance weighting. In particular, each hypothesis is assigned importance weight proportional to its data chunk size (see Eq. 3). The regularization parameter is fixed across different machines. The main contribution of the work is a consistency bound. In comparison to a previous bound on the divide & conquer algorithm (Li et al., arXiv 2019), this one does not require a constant number of machines (in my understanding of the related work section).

##### clarity
The paper is clear and easy to follow. I would say that the related work is fairly well covered and the contributions are appropriately placed in this regard.

##### quality & significance
I have a fundamental disagreement when it comes to the considered distributed setting. In particular, the bottleneck in learning with random Fourier features is not the size of the dataset but the number of features. The computational complexity is linear in the dataset size and cubic in the number of features. Moreover, there are examples of machine learning problems where it is required to use a huge number of features for satisfactory results (e.g., see Kernel Approximation Methods for Speech Recognition, May et al.). Thus, I do not see this direction as significant. After all, the algorithm improves over variable amounting to linear computational complexity. What would be interesting is to use different sets of random features on different machines and then aggregate on the master machine. In that way, one would be tackling the factor contributing to cubic computational complexity.

---

> ### Author Response · Authors · 2020-11-13
> **Response to Reviewer1**
>
> The direction of distributed setting in learning with random features is very significant, see below for reasons:
>
> (1) The total computational complexity of the learning with random features for KRR is $O(M^3+M^2|\mathcal{D}|)$, where requiring $O(M^3)$ to solve the inverse of $(\mathbf \Phi_{M,\mathcal{D}}\mathbf \Phi_{M,\mathcal{D}}^\mathrm{T}+\lambda \mathbf I)$ and $O(M^2|\mathcal{D}|)$ to solve the matrix multiplication $\mathbf \Phi_{M,\mathcal{D}} \mathbf \Phi_{M,\mathcal{D}}^\mathrm{T}$, $M$ is the size of random features, please see Section 2 in detail. In (Rudi \& Rosasco, 2017), they show that the optimal generalization bounds of KRR can be derived requiring only $O(\sqrt{|\mathcal{D}|})$ random features in the basic setting, as one can see that $M=\sqrt{|\mathcal{D}|}\ll|\mathcal{D}|$,  so the total computational complexity is $\mathcal{O}(M^2|\mathcal{D}|)$. Thus, the computational bottleneck in learning with random Fourier features to guarantee the optimal generalization performance not only depends on the size of random features, but also on the size of dataset. If we don't consider reducing the size of $\mathcal{D}$, the computational complexity is $|\mathcal{D}|^2$ in the basic setting, which is not suitable for large scale problems. Thus, how to reduce the size of $\mathcal{D}$ is a key problem to further improve the effectiveness for optimal learning rates;
>
> (2) Distributed learning is one of the most popular methods to reduce the size of dataset. The distributed learning bring the distributed error, but can decrease the variance of the model (See Proposition 1 in appendix or Zhang et al., 2013, 2015, or Li et al., 2019a). Thus, how to choose an appropriate number of partitions to trade off the distributed error and the variance to guarantee the optimal performance is a very interesting and significant direction. In this paper, we prove that the optimal learning rate in expectation is shown to achieve by relaxing the requirement on the number of partitions from $O(1)$ (Li et al., 2019a) to $O(|\mathcal{D}|^{0.5})$ in the basic setting (Theorem 1), so we can reduce the computational complexity from $\mathcal{O}(|\mathcal{D}|^2)$ (plain KRR with random features) to $\mathcal{O}(|\mathcal{D}|^{1.5})$. From (Rudi \& Rosasco, 2017; Rudi et al., 2018), we can further reduce the size of random features by generating in a data-dependent manner under some more stricter conditions (Theorem 3 in Rudi \& Rosasco, 2017). However, in this case, the item $|\mathcal{D}|$ has no change, and even become more dominate in total computational complexity. If combining our results (see Theorem 4,5,6 in appendix) with the data-dependent manner-based random features (Rudi \& Rosasco, 2017) under some more stricter conditions, we can reduce the size of random features and increase the partitions at the same time (see Remark 4 in appendix).
>
> (3) Moreover, when combining the distributed learning, random features and the preconditioned conjugate gradient (PCG), we can further speed up the KRR (see in Remark 3), which may open a path to reach the linear time complexity for optimal learning rate in the basic setting.
>
> Overall, the distributed setting for random features is a very interesting and significant direction for large scale kernel-based method. As commented by Reviewer 4, "the improvements are essential and would be helpful for the associated community".

---

### Author Response · Authors · 2020-11-22
**Paper Revision**

We have uploaded a new version of the paper that takes into account the comments from all reviewers.
The significant changes are as follows:
1) Remark 3, added in Section 4, which clarifies the significance of distributed learning for RF;
2) A major revision to improve our presentations as well as the citations,  added in Section 1;
3) More experiments on different datasets (minist, a8a, a6a, space-ca, cpusmall and abalone), added in APPENDIX  G.
4) More details of the selection of the optimal $\lambda$, added in Section 5.
5) More details of the gradient information of an operator representation in (Lin et al., 2020), added in Section 4.3.
6) The mention of our extention from the communication strategy in (Lin et al., 2020) to random features to avoid communicating local data among partition nodes, added at the beginning of Section 3.

Thanks to all reviewers for their constructive comments.

---

### Decision · Program_Chairs · 2021-01-07
**Final Decision**

**Decision:**

Accept (Poster)

**Comment:**

The focus of the submission is kernel ridge regression in the distributed setting. Particularly, the authors present optimal learning rates under this assumption both in expectation and in probability, while they relax previous restrictions on the number of partitions taken. The effectiveness of the approach is demonstrated in synthetic and real-world settings.

As summarized by the reviewers, the submission is well-organized and clearly written, the authors focus on an important problem, they present a fundamental theoretical contribution which also has clear practical impact. As such the submission could be of interest to the ICLR and ML community.